# Molecular pathway and structural mechanism of human oncochannel TRPV6 inhibition by the phytocannabinoid tetrahydrocannabivarin

Arthur Neuberger [1], Yury A. Trofimov [2], Maria V. Yelshanskaya[1], Jeffrey Khau [1], Kirill D. Nadezhdin [1], Lena S. Khosrof[1], Nikolay A. Krylov [2], Roman G. Efremov[2] & Alexander I. Sobolevsky [1] ✉

The calcium-selective oncochannel TRPV6 is an important driver of cell proliferation in human cancers. Despite increasing interest of pharmacological research in developing synthetic inhibitors of TRPV6, natural compounds acting at this channel have been largely neglected. On the other hand, pharmacokinetics of natural small-molecule antagonists optimized by nature throughout evolution endows these compounds with a medicinal potential to serve as potent and safe next-generation anti-cancer drugs. Here we report the structure of human TRPV6 in complex with tetrahydrocannabivarin (THCV), a natural cannabinoid inhibitor extracted from *Cannabis sativa*. We use cryo-electron microscopy combined with electrophysiology, calcium imaging, mutagenesis, and molecular dynamics simulations to identify THCV binding sites in the portals that connect the membrane environment surrounding the protein to the central cavity of the channel pore and to characterize the allosteric mechanism of TRPV6 inhibition. We also propose the molecular pathway taken by THCV to reach its binding site. Our study provides a foundation for the development of new TRPV6-targeting drugs.

For millennia, humans have been using *Cannabis sativa* preparations for medicinal purposes[1]. In modern history, however, decades of cannabis-associated stigma and legal restrictions in view of drug abuse concerns contributed to the delayed progress in much-needed research of the significant therapeutic values of various phytocannabinoids. Only recently, due to legalization and substantial mitigation of stigma, cannabis, its natural products and synthetic analogs, most of which have an obscure mechanism of action, have been reevaluated for therapeutic applications. Numerous diseases, such as cancer, anorexia, emesis, pain, inflammation, multiple sclerosis, neurodegenerative disorders (Parkinson's disease, Huntington's disease, Tourette's syndrome, Alzheimer's disease), epilepsy, glaucoma,

osteoporosis, schizophrenia, cardiovascular disorders, obesity, and metabolic syndrome-related disorders are being treated or have a potential to be treated by cannabinoid-based bioactive compounds[2]. In particular, therapeutic targeting of TRP channels by cannabinoids and their analogs has already been demonstrated for TRPV1–4, TRPA1, and TRPM8[3].

TRPV6, a representative of the vanilloid subfamily of transient receptor potential (TRP) channels, is the principal calcium uptake channel that captures diet-delivered calcium ions in the gut[4–6]. Dysregulation of this essential gatekeeper and master regulator of calcium uptake leads to disturbed calcium homeostasis. For example, knockout *Trpv6*[−/−] mice have been shown to exhibit defective absorption of

[1]Department of Biochemistry and Molecular Biophysics, Columbia University, New York, NY, USA. [2]Shemyakin-Ovchinnikov Institute of Bioorganic Chemistry, Russian Academy of Sciences, Moscow, Russia. ✉e-mail: as4005@cumc.columbia.edu

intestinal $Ca^{2+}$, increased urinary $Ca^{2+}$ excretion, decreased femoral bone mineral density, lower body weight, alopecia, dermatitis, and severely impaired male fertility[7–11]. Moreover, several human TRPV6 gene mutations have recently been linked to transient neonatal hyperparathyroidism, skeletal under-mineralization and dysplasia, hypercalciuria, chronic pancreatitis, various reproductive diseases, Pendred syndrome and Crohn's-like disease[12–24].

Given the central role of calcium in cancer development[25], it is not surprising that TRPV6 was shown to overexpress in some of the most aggressive human cancers, including breast, prostate, colon, ovarian, thyroid, endometrial cancers, and leukemia[25–32]. For instance, Peleg et al. found that TRPV6 overexpression contributes to colonic crypt hyperplasia in mice and colon cancer cell proliferation in humans[30]. Accordingly, suppression of TRPV6 expression was suggested to underlie cancer protective effects in the colon caused by high calcium diet[30]. Strikingly, in 93% of biopsies, the levels of TRPV6 were elevated to greater extent in invasive than in noninvasive tumor areas[33]. Moreover, the ancestral variant of this oncochannel has recently emerged as a driver of higher incidence, higher mortality, and more aggressive forms of cancer in people of African descent[34]. Drugs capable of regulating TRPV6 activity are therefore urgently needed.

While pharmacological research has mostly focused on synthetic inhibitors of TRPV6[35–38], natural compounds have largely been neglected. This occurs despite the substantial benefits of natural compounds including their pharmacokinetics that have been optimized by nature in the course of evolution[39]. Natural compounds therefore represent a much-needed foundation for the next-generation of small molecule inhibitors[39]. For instance, we have recently reported a structure of human TRPV6 in complex with the antagonist phytoestrogen genistein, a major isoflavone in nutritional soy, which causes channel closure and converts it into a novel two-fold symmetrical conformation[40]. Phytocannabinoid tetrahydrocannabivarin (THCV), a naturally occurring non-psychoactive analog of tetrahydrocannabinol found in *Cannabis sativa*, has also previously been shown to inhibit TRPV6[41]. Since THCV is a phytocannabinoid with low toxicity and side effects, it offers the possibility to target some of TRPV6-related diseases or help their prevention via THCV administration through nutritional diet.

Here we demonstrate that THCV acts as a micromolar-range inhibitor of TRPV6 and report the structure of human TRPV6 in complex with THCV. We employ cryo-electron microscopy (cryo-EM) combined with calcium imaging, electrophysiology, mutagenesis, and molecular dynamics (MD) simulations to understand the mechanism of TRPV6 inhibition by THCV. THCV binds in the transmembrane region of TRPV6, in a portal that connects the membrane environment surrounding the protein with the central cavity of the ion channel pore. Using MD simulations and cryo-EM, we also explore and reconstruct the molecular pathway, which is taken by THCV to reach the portal binding site. We propose an allosteric mechanism of hTRPV6 inhibition by THCV that has not been reported before and which lays an important foundation for the development of much-needed drugs targeting TRPV6.

## Results

### Functional characterization of TRPV6 inhibition by THCV
TRPV6 is a constitutively open channel[42–44]. In response to a −100 to 70 mV voltage ramp, we recorded a typical inward-rectified whole-cell patch-clamp current from HEK 293 cells expressing human TRPV6 (Fig. 1a). In the presence of 50 μM THCV, the current remained inwardly rectified but its amplitude was significantly reduced (at −70 mV, $I_{THCV}/I_{Control} = 0.123 \pm 0.029$, $n = 8$). We also monitored hTRPV6 inhibition by THCV using Fura-2 AM-based measurements of changes in intracellular $Ca^{2+}$. Changes in the fluorescence intensity ratio at 340 and 380 nm ($F_{340}/F_{380}$) evoked by addition of 10 mM $Ca^{2+}$ were measured after pre-incubation of hTRPV6-expressing HEK 293 cells with various

concentrations of THCV (Fig. 1b). THCV inhibited hTRPV6-mediated $Ca^{2+}$ uptake with the half-maximal inhibitory concentration, $IC_{50}$, of $15.4 \pm 2.3$ μM ($n = 8$, Fig. 1c).

### Cryo-EM analysis of TRPV6 in the presence of THCV
We employed cryo-EM to characterize TRPV6 inhibition by THCV structurally (Supplementary Figs. 1, 2). To avoid potential interference between binding of THCV and calmodulin (CaM), we used a human TRPV6 construct, hTRPV6-CtD, with truncated CaM binding site-containing C-terminus and wild type-like function[35,38,45,46]. Cryo-EM micrographs for hTRPV6-CtD reconstituted in cNW30 nanodiscs in the presence of 114 μM THCV showed evenly dispersed particles with diverse angular coverage (Supplementary Fig. 2a). Clearly visible secondary structure elements in 2D class averages supported high quality of the collected cryo-EM data (Supplementary Fig. 2b). The three-dimensional reconstruction of TRPV6$_{THCV}$ with four-fold rotational symmetry (C4) resulted in a cryo-EM map with the overall resolution of 2.79 Å (Fig. 1d–f, Supplementary Figs. 1, 2c–e, Supplementary Table 1). Consistent with the previously published structures of hTRPV6-CtD[35,38], the TRPV6$_{THCV}$ reconstruction contained no signs of CaM. For each subunit of the TRPV6$_{THCV}$ homotetramer, we built a model, including residues 27–638 and excluding the N-terminal (residues 1–26) and C-terminal (residues 639–666) regions that were not resolved in the cryo-EM density.

### Structure of TRPV6$_{THCV}$ and THCV binding sites
The TRPV6$_{THCV}$ structure (Fig. 2a, b) has a similar overall architecture as the previously determined TRPV6 structures[35,36,38,47]. TRPV6 is assembled of four subunits and contains a transmembrane domain (TMD) with the central ion channel pore and an intracellular skirt that is mostly built of ankyrin repeat domains connected to each other by three-stranded β-sheets, N-terminal helices, and C-terminal hooks. Amphipathic TRP helices run nearly parallel to the membrane and interact with both the TMD and the skirt. The TMD is composed of six transmembrane helices S1–S6 and a re-entrant pore loop (P-loop) between S5 and S6. A bundle of the first four transmembrane helices represents the S1–S4 domain which forms a voltage sensor in voltage-gated ion channels[48]. The pore domain of each subunit includes S5, P-loop, and S6, and is packed against the S1–S4 domain of the neighboring subunit in a domain-swapped arrangement[49].

The TMD of TRPV6$_{THCV}$ is surrounded by numerous auxiliary densities, the majority of which represent annular lipids (Fig. 1d–f). High-quality reconstruction of TRPV6$_{THCV}$ identifies three such densities per subunit as representing cholesteryl hemisuccinate (CHS), which was added during protein purification (see Methods). With the exception of two densities per subunit, other densities were modeled well by phosphatidylcholine (PC) molecules. The two exceptions (red in Fig. 1d, f) have not been seen in previous structures of TRPV6 and have the shape and size of THCV molecules (Supplementary Fig. 3a–d). The first one of the two corresponding THCV binding sites is formed by S5 and S6 of one subunit and S6 of the neighboring subunit and located in the portal connecting the pore and the membrane surrounding TMD (Fig. 2c, d). The second site is located next to the first one but it is more peripheral with respect to the pore, surrounded by lipids, including CHS molecules at the vanilloid site (at the interface of S5 and S6 of one subunit and S4 of the neighboring subunit) and at the interface of P-loop of one subunit and S6 of the neighboring subunit.

At both sites, putative THCV molecules do not form strong interactions with the surrounding residues (Fig. 2c, d, Supplementary Fig. 3e, f). At site 1, THCV makes a number of hydrophobic (Van der Waals) contacts with the residues surrounding the side portals. In contrast, the majority of weak interactions with THCV at site 2 are with lipids. Correspondingly, site 2 is unlikely to cause strong conformational changes in the protein and we propose that site 1 is the main inhibitory site of THCV.

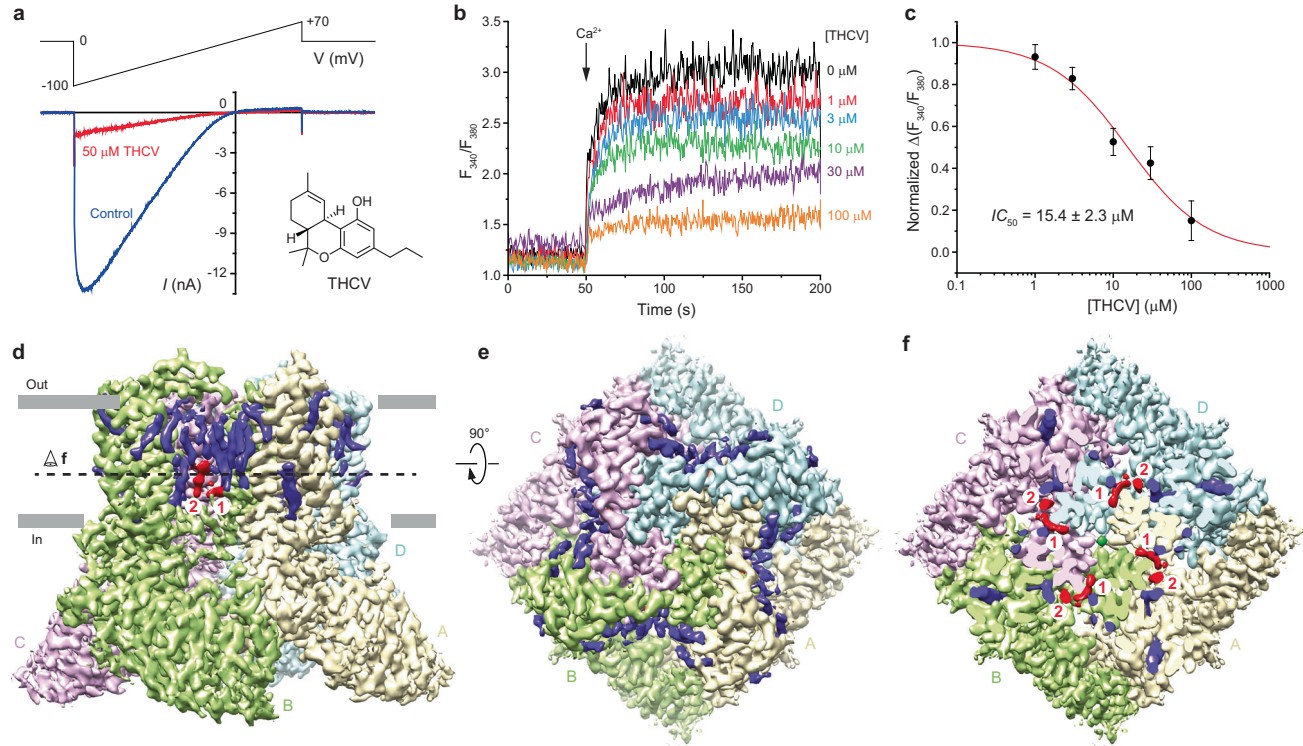

**Fig. 1 | Functional and cryo-EM characterization of TRPV6 inhibition by THCV.**
**a** Whole-cell patch-clamp currents recorded from HEK 293 cell expressing hTRPV6 in response to −100 to 70 mV voltage ramp in the absence (black) and presence (red) of 50 μM THCV. Examples are representatives of currents recorded from 8 different cells in two independent transfections. Recordings were made in the absence of $Ca^{2+}$, so the current was mainly carried by monovalent cations. The inset shows the chemical structure of THCV. **b** Representative ratiometric Fura-2 AM-based fluorescence measurements of changes in intracellular $Ca^{2+}$ for HEK 293 $GnTI^-$ cells expressing hTRPV6. The changes in the fluorescence intensity ratio at 340 and 380 nm ($F_{340}/F_{380}$) were monitored in response to application of 10 mM $Ca^{2+}$ (arrow) after pre-incubation of cells with various concentrations of THCV. The experiment was repeated independently eight times with similar results.

**c** Concentration–response curve for THCV inhibition of $Ca^{2+}$ uptake. The background subtracted changes in the fluorescence intensity ratio at 340 and 380 nm ($F_{340}/F_{380}$) evoked by the addition of 10 mM $Ca^{2+}$ after pre-incubation with various concentrations of THCV were normalized to the maximal change in $F_{340}/F_{380}$ after addition of 10 mM $Ca^{2+}$ in the absence of THCV. Data shown as mean ± SEM. Red curve through the data points is a fit with the logistic equation, with the mean ± SEM values of the half-maximum inhibitory concentration ($IC_{50}$), 15.4 ± 2.3 μM ($n = 8$ independent experiments). Source data are provided as a Source Data file. **d, e** 3D cryo-EM reconstruction of $TRPV6_{THCV}$ viewed from the side (**d**) or top (**e**), with subunits colored yellow, green, pink, and cyan. **f** Same cryo-EM density as in **e** but cut off along the dashed line in **d**. Putative densities for THCV and ions are shown in red and green.

## Probing of THCV binding site

To verify the identified sites 1 and 2, we made mutations of nearly all residues predicted to contribute to THCV binding (Supplementary Fig. 3e, f). Most residues were mutated to alanine, while alanine A561 and leucines L490 and L568 were mutated to the bulky tryptophan. We used fluorescence-detection size-exclusion chromatography (FSEC) to confirm expression and tetrameric assembly of the wild type and mutant channels produced in HEK 293 cells (Supplementary Fig. 4a–h) and tested their function using Fura-2 AM-based measurements of intracellular calcium. Based on the height of the FSEC tetrameric peak, the L490W mutant showed somewhat lower expression level (Supplementary Fig. 4b). Nevertheless, Fura-2 AM fluorescent signals were strong enough to make reliable measurements of intracellular calcium. Two mutants, C494W and L568W, did not show measurable calcium uptake. Confirming that these mutants were not functional, whole-cell recordings from HEK 293 cells expressing these mutants did not detect measurable TRPV6-mediated currents, while select examples of mutants that did show calcium uptake, conducted typical TRPV6-mediated currents (Supplementary Fig. 4i–l). As an alternative to the L568W mutation, we substituted leucine L568 with glutamine and the resulting L568Q mutant turned out to be functional.

Our calcium uptake measurements revealed that except for T567A, which showed a THCV concentration dependence similar to wild type (for example, at 30 μM THCV, the Δ($F_{340}/F_{380}$) values for the

mutant were not significantly different from wild type, $t$-Test, $P = 0.33$), all mutants were divided into two major groups: those that demonstrated a rightward shift in the THCV concentration dependence (M497A, A561W, L568Q) or weakening of the inhibition and those that showed a leftward shift in the concentration dependence (L490W, F493A, I564A) or increased potency of THCV correspondingly (Fig. 2e–g). Thus, replacement of the long hydrophobic side chain of M497 with the short side chain of alanine as well as substitution of hydrophobic leucine L568 with hydrophilic glutamine resulted in reduced potency of THCV (at 30 μM THCV, the Δ($F_{340}/F_{380}$) values were significantly different from wild type, $t$-Test, $P = 0.0081$ for M497A and $P = 0.0136$ for L568Q), likely due to the loss of hydrophobic interactions. On the other hand, weakening of THCV inhibition observed for the A561W mutant (at 30 μM THCV, the Δ($F_{340}/F_{380}$) values were significantly different from wild type, $t$-Test, $P = 0.0009$) most probably originated from more restricted access to the portals imposed by bulky side chain of tryptophan compared to the small side chain of alanine (Fig. 2c, d). On the contrary, smaller side chains of alanine in F493A and I564A likely made it easier for the THCV molecule to reach the deep portal site 1 or have created more room and opportunity for the inhibitor to form more optimal hydrophobic interactions with the surrounding protein, thus leading to an increased inhibitor potency (leftward shift in the THCV concentration dependence; at 30 μM THCV, the Δ($F_{340}/F_{380}$) values were significantly

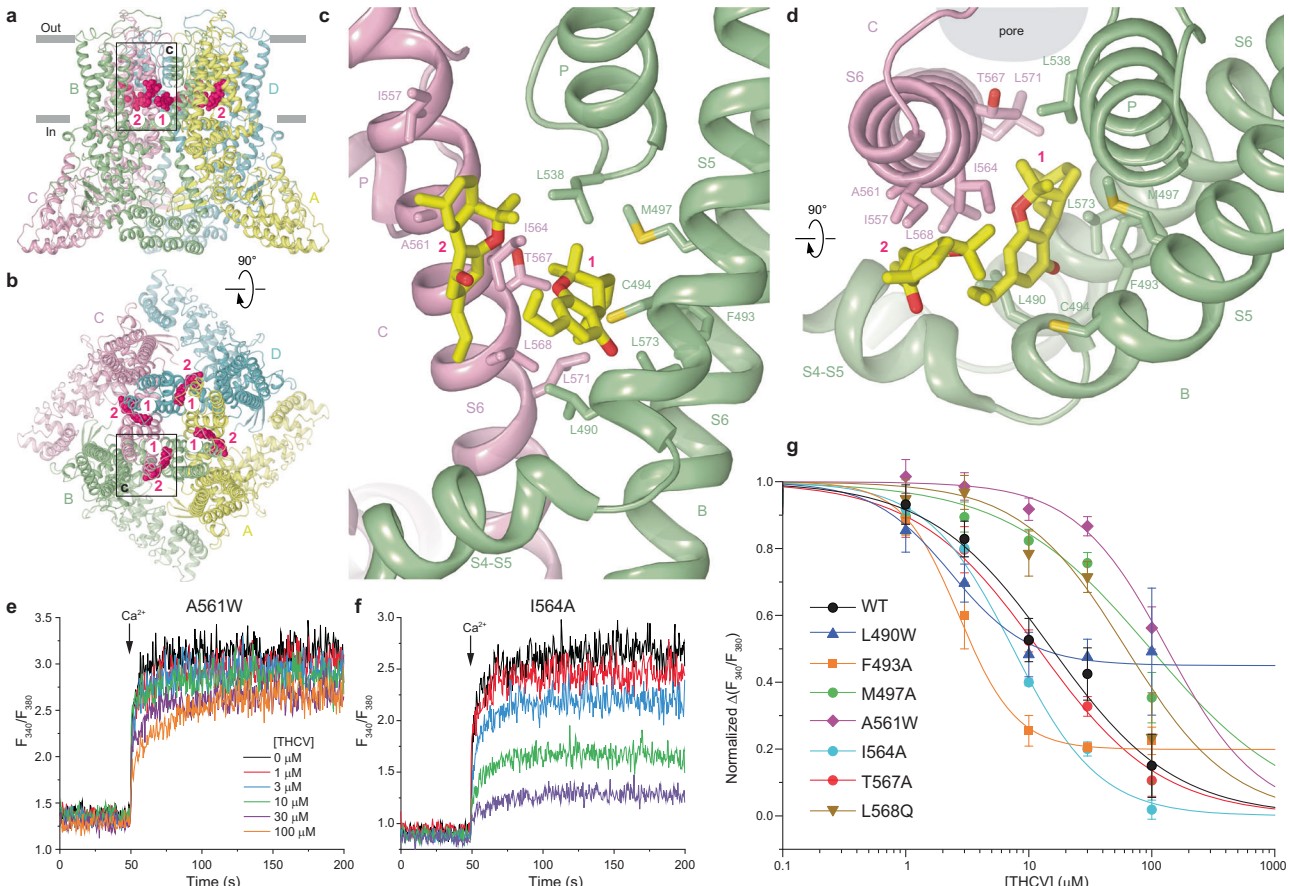

**Fig. 2 | Structure of TRPV6$_{THCV}$ and THCV binding sites. a, b** TRPV6$_{THCV}$ structure viewed from the side (**a**) or top (**b**), with subunits colored yellow, green, light pink, and cyan. THCV molecules are shown as space-filling models (hot pink) and the two types of sites are labeled. **c, d** Close-up views of the two THCV binding sites. **e, f** Representative ratiometric fluorescence measurements of changes in intracellular Ca$^{2+}$ for HEK 293 GnTI$^-$ cells expressing A561W (**e**) and I564A (**f**) mutant hTRPV6 channels. The changes in the fluorescence intensity ratio at 340 and 380 nm ($F_{340}/F_{380}$) were monitored in response to application of 10 mM Ca$^{2+}$ (arrow) after pre-incubation of cells with various concentrations of THCV. Each experiment was repeated independently four to eight times with similar results. **g** Concentration–response curves for inhibition of wild-type and mutant TRPV6

channels by THCV. The background subtracted changes in the fluorescence intensity ratio at 340 and 380 nm ($F_{340}/F_{380}$) evoked by the addition of 10 mM Ca$^{2+}$ after pre-incubation with various concentrations of THCV were normalized to the maximal change in $F_{340}/F_{380}$ after addition of 10 mM Ca$^{2+}$ in the absence of THCV. Data shown as mean ± SEM. Curves through the data points are fits with the logistic equation, with the mean ± SEM values of the half-maximum inhibitory concentration ($IC_{50}$), 15.4 ± 2.3 μM for WT ($n = 8$), 2.34 ± 0.32 μM for L490W ($n = 8$), 2.70 ± 0.23 μM for F493A ($n = 4$), 97.7 ± 45.2 μM for M497A ($n = 4$), 133.4 ± 23.6 μM for A561W ($n = 4$), 7.47 ± 0.41 μM for I564A ($n = 4$), 11.98 ± 0.70 μM for T567A ($n = 4$), and 62.1 ± 18.2 μM for L568Q ($n = 4$). Data for WT is the same as in Fig. 1c. Source data are provided as a Source data file.

different from wild type, *t*-Test, $P = 0.0429$ for F493A and $P = 0.0394$ for I564A).

Interestingly, replacement of leucine L490 that points towards sites 1 and 2 with the bulky tryptophan produces a dual effect: it increases THCV potency, likely due to increased number of hydrophobic interactions with the indole ring in one rotameric conformation, and the same time makes the inhibition ~50% incomplete, probably because the indole ring in an alternative rotameric conformation fills up the portal space and prevents binding of THCV (Fig. 2g; while at 30 μM THCV, the Δ($F_{340}/F_{380}$) values for L490W were not significantly different from wild type, *t*-Test, $P = 0.479$, they were significantly different at 3 μM THCV, *t*-Test, $P = 0.0185$, and 100 μM THCV, *t*-Test, $P = 0.0271$). In this regard, a small ~20% fraction of uninhibited F493A channels at high THCV concentration may originate from an alternative positioning of the THCV molecule in the roomier site 1, where it no longer causes conformational changes that lead to channel closure (see the next section). However, given high values of basal calcium signals in experiments with L490W (see Source Data), their results should be interpreted with caution. For select mutants that showed pronounced leftward (L490W and I564A) and rightward (M497A and A561W) shifts in $IC_{50}$ for THCV, we confirmed the ability to

conduct currents in whole-cell patch-clamp experiments (Supplementary Fig. 4i–l), lack of dramatic changes in the concentration dependencies of calcium activation (Supplementary Fig. 4m) and econazole inhibition (Supplementary Fig. 4n), and complete block of the L490W mutant by 100 μM Gd$^{3+}$ (Supplementary Fig. 4o), suggesting that functional properties other than THCV inhibitory potency were not strongly affected by the corresponding mutations. Interestingly, the slope of the concentration–response curve for econazole inhibition of M497A was less steep compared to wild type and other mutants (Supplementary Fig. 4n). Given that M497 is located on the opposite site of the S5 helix, close to W495, which makes a direct contact with the econazole molecule, it is possible that this mutation weakens the cooperativity that normally exists between four econazole binding sites (one per TRPV6 subunit). Overall, however, the changes in the concentration dependence of calcium uptake inhibition by THCV observed in mutant channels compared to wild type (Fig. 2g) strongly support THCV binding to sites 1 and 2.

## Closed pore of TRPV6$_{THCV}$

In contrast to the apo-state structures of hTRPV6, which have an open conducting pore[35,38,47], the pore in TRPV6$_{THCV}$ is in a fully closed

conformation, hydrophobically sealed by the side chains of residues L574 and M578 (Fig. 3a). Compared to the open-state structure with the π bulge in the middle of S6, the entire S6 in TRPV6$_{THCV}$ is α-helical, typical for the closed-state structures of TRPV6, including structures in complex with inhibitors ruthenium red (RR) and econazole[35,36,38,46,47,49] (Fig. 3b). Comparison of the open-state hTRPV6 structure to the structure solved in the presence of THCV demonstrates that the open-to-closed state transformation of the lower pore includes a ~100° rotation and bending away from the pore axis of the intracellular part of S6, consistent with previous observations[36,38,47]. This drastic conformational change in S6 starts below the gating hinge alanine A566, without altering the upper pore of hTRPV6 and thereby leaving the selectivity filter unaffected.

Within the hTRPV6 regions previously implicated in gating, two hydrogen bonds per subunit stabilize the open state of the channel: one bond between Q473 in the S4–S5 elbow and R589 in the TRP helix and another one between D489 in the S5 helix and T581 in the S6 helix[47]. Neither interaction is present in the closed-state structure of hTRPV6 in complex with THCV, suggesting that binding of THCV leads to disruption of these bonds. The closed pore of TRPV6$_{THCV}$ therefore confirms the inhibitory action of THCV. To further validate that this inhibitory action occurs through binding of THCV in the ion channel side portals and to provide an insight into the molecular mechanism of this inhibition, we performed molecular dynamics (MD) simulations.

### Orientation of THCV in site 1 probed with MD simulations

There are two possible orientations of THCV at site 1: (1) with the tricyclic ring of THCV oriented towards the central pore (forward orientation, Fig. 4a) and (2) with the tricyclic ring oriented towards the membrane (backward orientation, Fig. 4b). Since the cryo-EM map provides no clear density for the acyl chain of THCV due to its flexibility, both orientations are feasible. To determine a more favorable positioning of the ligand, we performed two sets of MD simulations, one for each orientation. In each set, we carried out four simulations, with the initial orientations of the ligand differed by 90° rotations around the long THCV axis. As a measure of ligand stability in the binding pocket, we calculated the average spatial distribution of the center of the THCV molecule determined as a center of mass of heavy atoms over the two central rings of THCV. The ligand center appeared to overlap well with the experimental cryo-EM density for THCV in the forward orientation (Fig. 4a), while it was shifted towards the central pore or membrane environment for THCV in the backward orientation (Fig. 4b). Compared to the latter, the forward orientation also showed a smaller displacement of the ligand center relative to the cryo-EM model over the simulation time (Fig. 4c). According to the MD data, the only specific interaction that can stabilize the forward orientation of THCV is the hydrogen bond between the hydroxyl group of THCV and the carbonyl oxygen of L490 (Fig. 4a). This bond helps to create and maintain a kink in the S5 helix and prevents bond formation between the adjacent D489 and T581 in S6 that stabilizes the open state of the channel (Supplementary Fig. 5a). By contrast, formation of the hydrogen bond between THCV and L490 in the backward orientation in our simulations leads to dissociation of the ligand from the binding pocket.

### THCV pathway to site 1 probed by constant velocity steered MD simulations

Since conventional MD simulations are too slow to model ligand binding and dissociation in such complex systems as TRPV6$_{THCV}$, we pulled the forward-oriented THCVs out of the binding pockets using the constant velocity steered MD simulations[50]. Four independently pulled ligands, one per subunit of TRPV6 tetramer, revealed two possible routes of leaving their binding sites. The first one, "horizontal", runs along the S4–S5-linker helix through a groove formed by the side chains of L490 and L568 to the region of the THCV site 2 (Fig. 4d). The second route, "vertical", runs along the pore helix substituting the adjacent annular CHS (lipid 803 in the cryo-EM model, Supplementary Fig. 5b).

To determine the energy landscapes for both pathways, we performed the umbrella sampling MD simulations[51] and calculated the potential of mean force (PMF) along the pulling trajectories. In both cases, the THCV position at the portal site was stabilized by the hydrogen bond between the ligand and L490 and corresponded to the energy minimum along the selected pathway (M1 region in Fig. 4e and Supplementary Fig. 5c). There was another preferable position of the ligand along the vertical route, where THCV was involved in a π-

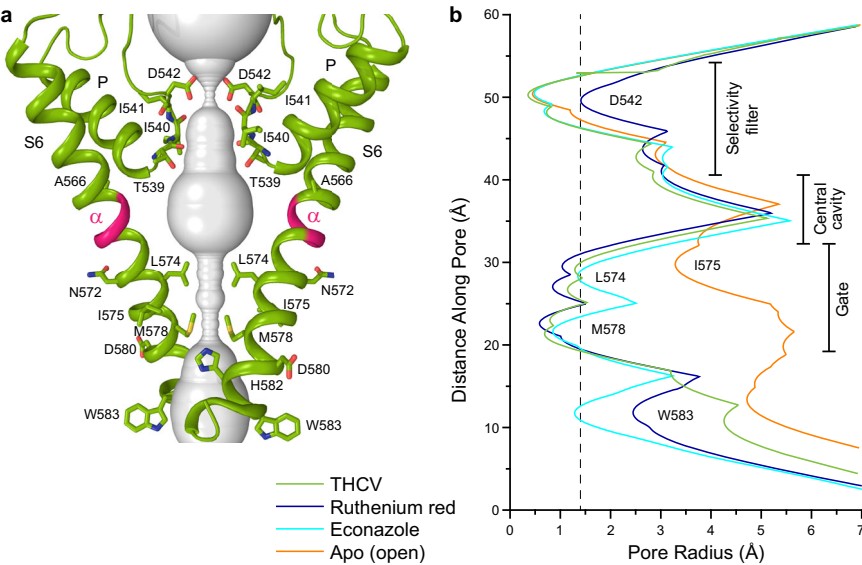

**Fig. 3 | TRPV6$_{THCV}$ Pore. a** Pore-forming domain in TRPV6$_{THCV}$ with the residues contributing to pore lining in the THCV-bound (TRPV6$_{THCV}$), ruthenium red-bound (TRPV6$_{RR}$), econazole-bound (TRPV6$_{Eco}$), and apo-state (TRPV6$_{Open}$) structures shown as sticks. Only two of four subunits are shown, with the front and back subunits omitted for clarity. The pore profile is shown as a space-filling model (gray). The region that undergoes α-to-π transition in S6 is highlighted in pink. **b** Pore radius for TRPV6$_{THCV}$ (green), TRPV6$_{RR}$ (blue, PDB ID: 7S8B), TRPV6$_{Eco}$ (cyan, PDB ID: 7S8C) and TRPV6$_{Open}$ (orange, PDB ID: 7S89) calculated using HOLE. The vertical dashed line denotes the radius of a water molecule, 1.4 Å.

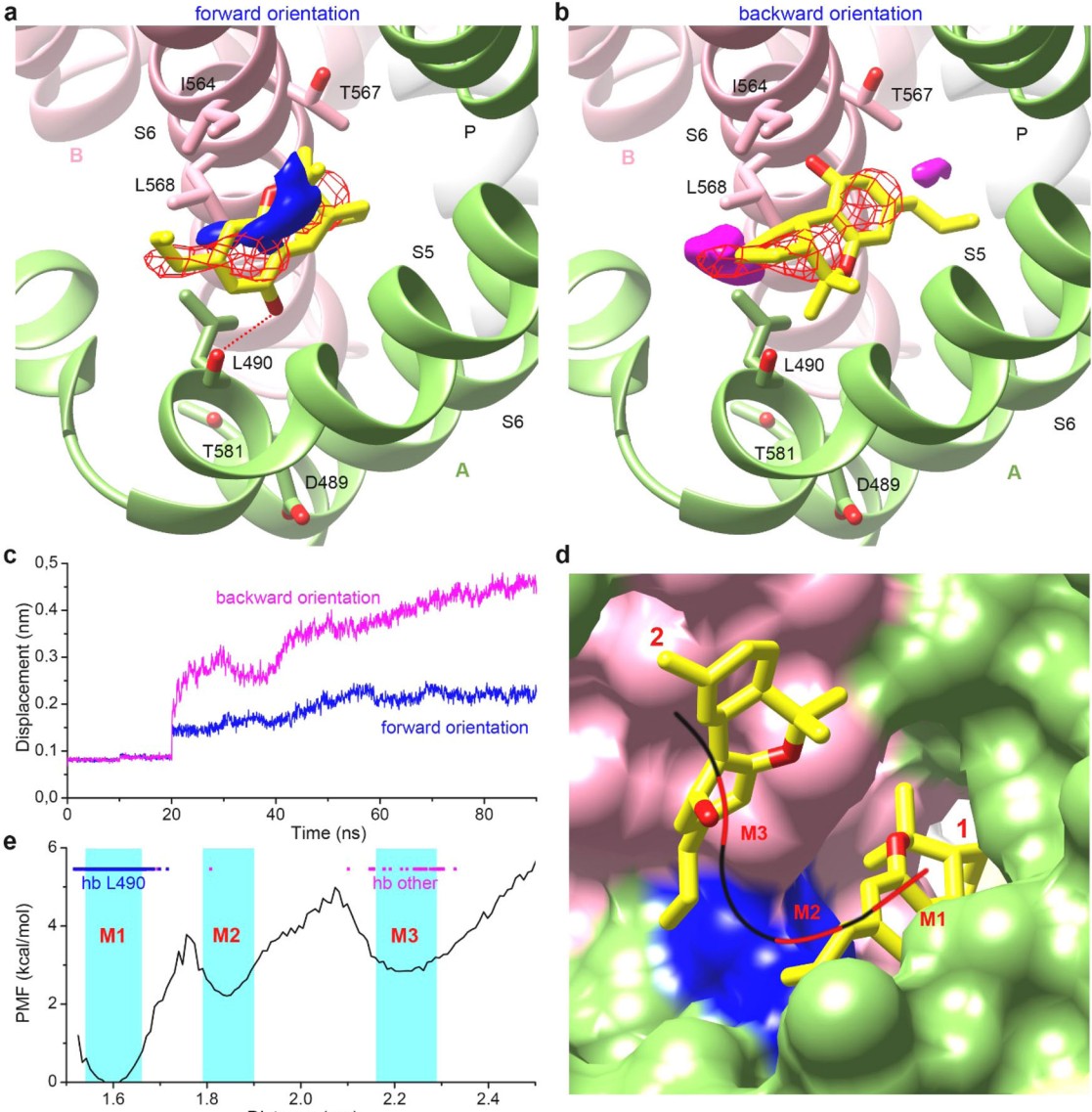

**Fig. 4 | MD simulation of THCV binding to TRPV6. a**, **b** THCV binding to site 1 with the tricyclic ring oriented towards the central pore (forward orientation, **a**) and towards the membrane (backward orientation, **b**). Cryo-EM model of protein is shown as green (subunit A) and pink (subunit B) cartoon, while THCV is shown as yellow sticks. Distributions of the center of THCV molecule modeled in the forward and backward orientations are shown as blue and pink surfaces, respectively. Red mesh represents cryo-EM non-protein density. The hydrogen bond between THCV and L490 is shown as a red dotted line. **c** Time dependence of averaged displacement of the center of THCV molecule modeled in the forward (blue) and backward (pink) orientations relative to the position in the cryo-EM model (the

ligands positions were fixed during the first 20 ns). **d** THCV horizontal pathway is shown as a black curve, with energy minima along the pathway indicated by red segments M1, M2, and M3. Surface of subunit A, subunit B and a groove along the helices S4–S5 contributed by L490 and L568 is colored green, pink and blue, respectively. **e** Potential of mean force calculated along the horizontal pathway as a function of the distance between the centers of the THCV molecule and the pore axis, with the energy minima M1, M2 and M3 highlighted in cyan. Blue and pink lines indicate hydrogen bonds between THCV and L490 and between THCV and other residues, respectively.

stacking interaction with F534 near the pore helix (M3 region in Supplementary Fig. 5c). However, the transition from M1 to M3 that followed the vertical route was associated with overcoming the energy barrier of 6–7 kcal/mol caused by significant protein-ligand clashes. In contrast, the horizontal route ran through the M1-M2-M3 energy minima separated by energy barriers of only 3–4 kcal/mol (Fig. 4e).

To check the ligand binding in the same pockets of the TRPV6 open state, we performed one more conventional MD simulation with the forward-oriented THCVs (like in the cryo-EM model) embedded in site 1 of apo TRPV6 (PDB ID 7S88)[38]. The simulation showed that THCV molecules lost the hydrogen bond with L490 and left the pockets right after removing the positional restraints applied to the ligands. Three out of four ligands moved out of the pockets along the horizontal

route, now formed by L490 and L569 (Supplementary Fig. 5a). We therefore concluded that the horizontal route connecting THCV sites 1 and 2 is the most likely pathway for the ligand to approach or leave its portal binding pocket both in the TRPV6 inhibited and open states.

Consistent with the nonpolar character of the THCV molecule, the horizontal route also has pronounced hydrophobic properties. This reciprocity can be clearly illustrated by using the molecular hydrophobicity potential (MHP) method[52,53]. Thus, with the exception of the polar OH group, which is capable of forming hydrogen bonds with the protein, the surface of THCV is hydrophobic and characterized by high MHP values (Supplementary Fig. 5d). A similar pattern is observed for the surrounding protein surface (Supplementary Fig. 5e). Such a complementarity in physicochemical properties between the protein

and the ligand (contacts of their hydrophobic surfaces are energetically favorable) facilitates the process of ligand migration between its binding sites in TRPV6.

To rationalize the effects of mutation L490W in THCV site 1 (Fig. 2g), we performed MD simulations of the corresponding mutant TRPV6$_{THCV}$ channels with the forward-oriented THCV molecules embedded in site 1 like in cryo-EM model. MD simulations showed that the indole ring of tryptophan W490 can adapt two possible conformations, with the side chain oriented inward and outward the pocket (Supplementary Fig. 6a). In both conformations, W490 forms a parallel π-stacking interaction with the tricyclic ring of THCV (Supplementary Fig. 6b, c). The improved stability of the hydrogen bond between THCV and W490 as well as additional π-stacking interaction (Supplementary Fig. 6d) explain the increased affinity of the ligand observed for L490W in calcium uptake experiments (Fig. 2g). The significantly reduced ability of THCV to inhibit calcium entry through the L490W mutant is likely due to limited access of the ligand to the portal site because of the block of the horizontal route by the bulky side chain of tryptophan. Our MD analysis, therefore, provides a rationale for the results observed in calcium uptake experiments and reinforces the importance of site 1 for the mechanism of TRPV6 inhibition by THCV.

### Mechanism of TRPV6 inhibition by THCV

To accommodate binding of THCV to site 1, the surrounding side chains must rearrange to make space between S5 and S6 of the neighboring subunits (Fig. 5a). These residues include F493 and M497 on S5 that mainly change their side-chain conformation. However, the S4–S5-linker residues N-terminal to F493 not only alter their side chain positions but also move the backbone, causing a rotation of the C-terminal part of the S4–S5-helix. Similarly, while the N-terminal part of S6 and the pore loop remain essentially intact, the S6 residues C-terminal to the gating hinge alanine A566 move their side chains together with the protein backbone. This movement represents a

~100° rotation of the C-terminal portion of S6, with the side chains of T567, L568, and L571 making contacts with THCV. The rotation introduces a π-to-α transition in S6, making it entirely α-helical, causes a ~2 helical-turn shortening of S6, a ~2 helical-turn elongation of the TRP helix and repositions M578 so that its side chain points towards the channel pore center (Fig. 5b, c). In this position, M578 side chains hydrophobically seal the pore for water and ion conductance, finalizing the open-to-closed state transition (Fig. 5d, e). THCV therefore acts as a cog that inserts itself in a cogwheel mechanism of TRPV6 channel and ensures its transition from the open to closed state.

## Discussion

Our study shows that the main inhibitory sites of THCV in human TRPV6 are located in the side portals connecting the channel pore to the membrane environment. This location is different from binding sites of TRPV6 ligands that have so far been characterized structurally (Supplementary Fig. 7a, b). Of 14 ligand-binding sites identified in TRPV channels[54], this deep portal site was previously shown to bind the agonist cannabidiol (CBD) in TRPV2[55,56] (Supplementary Fig. 7c, d) and local anesthetic dyclonine in TRPV3[57] (Supplementary Fig. 7e, f). Dyclonine, however, inserts itself deeper into the portal site and sticks out into the channel pore, thus creating a direct barrier for conductance of water and ions. CBD, on the contrary, does not stick out to the pore but allosterically causes the opposite to THCV action—channel opening. Interestingly, CBD and other cannabinoids were previously demonstrated to be ineffective as antagonists of both TRPV5 and TRPV6, at least at the tested doses[41]. Compared to dyclonine, THCV appears to have a unique allosteric mechanism of inhibition among TRPV channels. According to this mechanism, THCV stabilizes a closed-state conformation of TRPV6 by acting like a molecular cog inserted into the cogwheel mechanism of the channel to alter its gating state.

Our MD simulations suggest the pathway that THCV uses to reach the deep portal binding site 1. Of the two analyzed routes, the

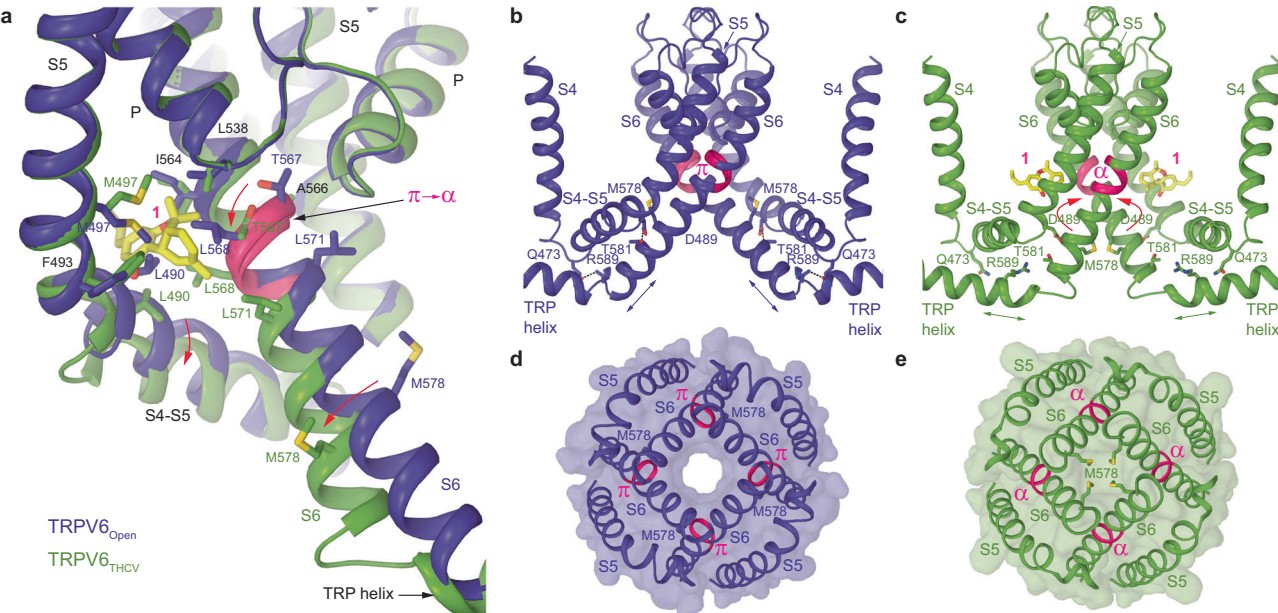

**Fig. 5 | Conformational changes and mechanism of TRPV6 inhibition by THCV.**
**a** Superposition of TRPV6$_{THCV}$ (green) and apo-state TRPV6$_{Open}$ (purple, PDB ID: 7S89) structures. THCV molecule at site 1 (yellow) as well as residues contributing to THCV binding and forming the gate (M578) are shown as sticks. Conformational changes induced by THCV binding are indicated by red arrows. The region that undergoes the π-to-α transition in S6 upon THCV binding is highlighted (pink). **b, c** Core of the transmembrane domain in TRPV6$_{Open}$ (**b**) and TRPV6$_{THCV}$ (**c**), with

elongated S6 and TRP helices indicated by double arrows, respectively. Open-state stabilizing interactions between Q473 and R589 as well as D489 and T581 are shown as dotted lines. Only two of four subunits are shown, with the front and back subunits removed for clarity. **d, e** Intracellular views of the gate region in TRPV6$_{Open}$ (**d**) and TRPV6$_{THCV}$ (**e**), with the surface shown in the corresponding color and M578 labeled.

horizontal one appears to be more plausible. Interestingly, this horizontal pathway is the one that runs through THCV site 2, which was also identified structurally. We hypothesize that site 2 is an intermediate stop for THCV that travels through the membrane to reach site 1. It likely has an increased affinity to THCV compared to other locations throughout the horizontal route, causing dwelling of THCV at site 2, which reveals itself as the corresponding density in the cryo-EM map. Interestingly, mutations of residues coordinating THCV at site 1 to bulky hydrophobic residues allow additional π-stacking interactions with the THCV tricyclic ring structure and result in the increased inhibitor affinity towards the mutant channels. This also suggests a possibility for the development of more potent THCV-inspired inhibitors with better selectivity, distribution patterns, and pharmacokinetics to fill a much-needed medicinal niche for fine-tuning TRPV6-mediated activity in numerous diseases, including cancer.

THCV inhibition of TRPV5, a close relative of TRPV6 (75% overall amino acid sequence identity), was shown to slow the progression of joint destruction in a rat model for osteoarthritis, highlighting a new potential therapeutic application of THCV for treatment of osteoarthritis[58]. Given the high degree of structural similarity between TRPV5 and TRPV6, it is likely that THCV uses the same mechanism of TRPV5 inhibition as for TRPV6 described in this study. Indeed, sequence alignment of TRPV5 and TRPV6 (Supplementary Fig. 8) reveals that 10 out of 11 residues coordinating THCV at site 1 in TRPV6 (Supplementary Fig. 3e) are the same in TRPV5. Given the aromatic character of the side chain of F574 and important contribution of the introduced π-stacking interactions in increased affinity of THCV to TRPV6 mutants predicted by MD simulations (Fig. 4, Supplementary Fig. 6), we expect that THCV binding to TRPV5 may show increased potency and specificity compared to TRPV6.

## Methods

### Construct
C-terminally truncated human TRPV6 (hTRPV6-CtD, residues 1–666 of wild-type channel) used for cryo-EM was cloned into a pEG BacMam vector with a C-terminal thrombin cleavage site followed by a streptavidin affinity tag (WSHPQFEK). For Fura-2 AM measurements, point mutations in wild-type human TRPV6 were introduced using standard molecular biology techniques. For FSEC experiments, full-length wild-type or mutant human TRPV6 was cloned into a pEG BacMam vector with a C-terminal thrombin cleavage site followed by a streptavidin affinity tag (WSHPQFEK) and enhanced green fluorescent protein (eGFP).

### Expression and purification
hTRPV6-CtD was expressed and purified based on our previously established protocols[38,40,59–63]. Bacmids and baculoviruses were produced using standard procedures. Baculovirus was made in Sf9 cells for ~72 h (Thermo Fisher Scientific, mycoplasma test negative, GIBCO #12659017) and was added to suspension-adapted HEK 293S cells lacking N-acetyl-glucosaminyltransferase I (GnTI⁻, mycoplasma test negative, ATCC #CRL-3022) that were maintained in Freestyle 293 media (Gibco-Life Technologies #12338-018) supplemented with 2% FBS at 37 °C and 5% CO₂. Twenty-four hours after transduction, 10 mM sodium butyrate was added to enhance protein expression, and the temperature was reduced to 30 °C. Seventy-two hours after transduction, cells were harvested by centrifugation at $5471 \times g$ for 15 min using a Sorvall Evolution RC centrifuge (Thermo Fisher Scientific), washed in phosphate-buffered saline pH 8.0, and pelleted by centrifugation at $3202 \times g$ for 10 min using an Eppendorf 5810 centrifuge. The cell pellet was solubilized under constant stirring for 2 h at 4 °C in ice-cold lysis buffer containing 1% (w/v) n-dodecyl β-D-maltoside, 0.1% (w/v) CHS, 20 mM Tris-HCl pH 8.0, 150 mM NaCl, 0.8 μM aprotinin, 4.3 μM leupeptin, 2 μM pepstatin A, 1 mM

phenylmethylsulfonyl fluoride, and 1 mM β-mercaptoethanol (βME). The non-solubilized material was pelleted in the Eppendorf 5810 centrifuge at $3202 \times g$ and 4 °C for 10 min. The supernatant was subjected to ultracentrifugation in a Beckman Coulter ultracentrifuge using a Beckman Coulter Type 45Ti rotor at $186,000 \times g$ and 4 °C for 1 h to further clean up the solubilized protein. The supernatant was added to strep resin and rotated for 1 h at 4 °C. The resin was washed with 10 column volumes of wash buffer containing 20 mM Tris-HCl pH 8.0, 150 mM NaCl, 1 mM βME, 0.01% (w/v) GDN, and 0.001% (w/v) CHS, and the protein was eluted with the same buffer supplemented with 2.5 mM D-desthiobiotin. The eluted protein was concentrated using a 100 kDa NMWL centrifugal filter (MilliporeSigma Amicon) to 0.5 ml and then centrifuged in a Sorvall MTX 150 Micro-Ultracentrifuge (Thermo Fisher Scientific) using an S100AT4 rotor for 30 min at $66,000 \times g$ and 4 °C before injection into a size-exclusion chromatography (SEC) column. All constructs were further purified using a Superose™ 6 10/300 GL SEC column attached to an AKTA FPLC (GE Healthcare) and equilibrated in 150 mM NaCl, 20 mM Tris-HCl pH 8.0, 1 mM βME, 0.01% GDN, and 0.001% CHS. The tetrameric peak fractions were pooled and concentrated using 100 kDa NMWL centrifugal filter to 3.5 mg/ml.

Circularized cNW30 nanodiscs were prepared according to the standard protocol[48] and stored (~2–3 mg/ml) before usage at −80 °C in 20 mM Tris-HCl (pH 8.0) and 150 mM NaCl. For nanodisc reconstitution, the purified protein was mixed with circularized cNW30 nanodiscs and soybean lipids (Soy polar extract, Avanti Polar Lipids, USA) at a molar ratio of 1:3:166 (TRPV6 monomer:cNW30:lipid). The lipids were dissolved to a concentration of 100 mg/ml in 150 mM NaCl and 20 mM Tris-HCl (pH 8.0) and subjected to 5–10 cycles of freezing in liquid nitrogen and thawing in a water bath sonicator. The nanodisc mixture (500 μl) was rocked at room temperature for 1 h. Subsequently, 20 mg of Bio-beads SM2 (Bio-Rad), pre-wet in a buffer containing 20 mM Tris-HCl pH 8.0, 150 mM NaCl and 1 mM βME, were added to the nanodisc mixture, which was then rotated for 1 h at 4 °C. Additional 20 mg of Bio-beads SM2 were added, and the resulting mixture was rotated at 4 °C for another ~14–20 h. Bio-beads SM2 were then removed by pipetting and nanodisc-reconstituted hTRPV6-CtD was purified from empty nanodiscs by SEC using Superose™ 6 10/300 GL SEC column equilibrated in 150 mM NaCl, 20 mM Tris-HCl (pH 8.0), and 1 mM βME. Fractions of nanodisc-reconstituted hTRPV6-CtD were pooled and concentrated using 100 kDa NMWL centrifugal filter to 2.58 mg/ml. 114 μM THCV (from a 5 mM stock in 100% DMSO) was added to the TRPV6 monomer:cNW30:lipid mixture and the specimen was incubated for 1 h at room temperature before grid preparation.

### Cryo-EM sample preparation and data collection
UltrAuFoil R 1.2/1.3, Au 300 grids were used for plunge-freezing. Prior to sample application, grids were plasma treated in a PELCO easiGlow glow discharge cleaning system (0.39 mBar, 15 mA, "glow" 25 s, "hold" 10 s). A Mark IV Vitrobot (Thermo Fisher Scientific) set to 100% humidity at 4 °C was used to plunge-freeze the grids in liquid ethane after applying 3 μl of protein sample to their gold-coated side using a blot time of 5 s, a blot force of 5, and a wait time of 15 s. The grids were stored in liquid nitrogen before imaging.

Images of frozen-hydrated particles of cNW30-reconstituted hTRPV6₍THCV₎ were collected on a Titan Krios transmission electron microscope (Thermo Fisher Scientific) operating at 300 kV and equipped with a post-column GIF Quantum energy filter and a Gatan K3 Summit direct electron detection camera (Gatan, Pleasanton, CA, USA) using SerialEM 4.0. 6699 micrographs were collected in super-resolution counting mode with raw image pixel size of 0.4125 Å across the defocus range of −0.8 to −2.0 μm. The total dose of ~60 e⁻Å⁻² was attained by using the dose rate of ~16 e⁻pixel⁻¹s⁻¹ across 50 frames during the 2.0-s exposure time.

## Image processing and 3D reconstruction

Data were processed in cryoSPARC[64]. Movie frames were aligned using the patch motion correction. Contrast transfer function (CTF) estimation was performed on non-dose-weighted micrographs using the patch CTF estimation. Subsequent data processing was done on dose-weighted micrographs. Following CTF estimation, micrographs were manually inspected and those with outliers in defocus values, ice thickness, and astigmatism as well as micrographs with lower predicted CTF-correlated resolution (higher than 5 Å) were excluded from further processing (individually assessed for each parameter relative to the overall distribution). After several rounds of selection through 2D classification, particles were further 3D classified (heterogeneous refinement) into four classes. Particles representing the best class were re-extracted without binning (256-pixel box size) and further 3D classified. The final set of 213,249 particles for hTRPV6$_{THCV}$ representing the best class was subjected to homogenous refinement. The reported resolution of 2.79 Å was estimated using the gold standard Fourier shell correlation (GSFSC). The local resolution was calculated with the resolution range estimated using the FSC = 0.143 criterion. Cryo-EM density visualization was done in UCSF Chimera[51] and UCSF ChimeraX[65].

## Model building

To build the model of TRPV6$_{THCV}$ in Coot[66], we used the previously published cryo-EM structure of closed-state TRPV6[38] (PDB ID: 7S8C) as a guide. The model was tested for overfitting by shifting its coordinates by 0.5 Å (using Shake) in Phenix[67], refining the shaken model against the corresponding unfiltered half map, and generating densities from the resulting models in UCSF Chimera. TRPV6$_{THCV}$ and other structures were visualized and figures were prepared in UCSF Chimera, UCSF ChimeraX[65], and Pymol[68]. The pore radius was calculated using HOLE[69].

## MD simulations

To determine a suitable THCV structure for MD simulations, we first explored the conformational ensemble of this ligand using quantum chemical calculations. Since the most significant changes in the THCV conformation are associated with the rotation around the $C_{17}$-$C_{04}$ bond (see atom labels in Supplementary Fig. 3e, f), the dihedral angle $C_{14}$-$C_{17}$-$C_{04}$-$C_{03}$ was scanned to determine possible energy minima. As a result, two stable states were found in the energy landscape, with the dihedral angle values of 170° (conformation 1) and 91° (conformation 2). Equilibrium geometry (minimum energy conformation) of the isolated THCV and energy profiles for the states generated by means of the dihedral drive protocol were calculated in vacuo using the Hartree-Fock method with the 6-31G(d) basis in GAUSSIAN09, Revision A.01 software[70]. Geometry optimization for the calculated models was performed without restraints. The Charmm36[71,72] topology for THCV was created by providing the possibility of spontaneous conformational transitions between THCV conformations 1 and 2. The topology was successfully tested via MD simulations of the isolated THCV in vacuo. Since the cryo-EM model of THCV in the pocket is very close to conformation 2, we used this conformation in subsequent MD simulations with the channel.

Structural model of TRPV6$_{THCV}$ (residues 27–638) was inserted into a hydrated lipid bilayer with the molecular composition of 50% palmitoyloleoylphosphatidylcholine (POPC), 25% palmitoyloleoylphosphatidylethanolamine (POPE), and 25% cholesterol (about 900 molecules in the membrane). $Na^+$ and $Cl^-$ ions were added to ensure zero net charge at 0.15 M ionic concentration. Four THCV molecules were embedded with the same orientation into the portal pockets of the protein. Eight replicas of the system with the wild-type (WT) protein and different orientations of THCV (see text and Supplementary Table 2 for details), one replica with the L490W mutation and one replica of the system with THCV molecules embedded into the portal pockets of the open TRPV6[38] (PDB ID 7S88) were constructed this way.

The simulated systems were first equilibrated in several stages: $5 \times 10^3$ steps of steepest descent minimization followed by heating from 5 to 310 K during a 100-ps MD-run, then 10 ns of MD run with fixed positions of the protein and THCV heavy atoms, 10 ns of MD with fixed positions of the protein backbone and THCV heavy atoms, 20 ns of MD with fixed positions of the protein $C_\alpha$ atoms to permit membrane and THCV relaxation after insertion. Then, 50 ns MD-production runs were carried out for the WT protein and open TRPV6 systems, 200 ns MD-production runs were carried out for the mutated and one of the WT systems with THCV oriented like in the cryo-EM model.

MD simulations were carried out using GROMACS 2021.4 package[73], CHARMM36 force field[74–79] and the TIP3P water model[80]. Simulations were carried out with an integration time of 2 fs, constrained hydrogen-containing bond lengths by the LINCS algorithm[81], imposed 3D periodic boundary conditions, constant temperature (310 K) and pressure (1 bar). Cutoff distance of 1.2 nm was used for evaluation of nonbonded interactions and the particle-mesh Ewald method[82] employed for treatment of long-range electrostatics.

The WT system with forward-oriented THCV like in the cryo-EM model was used for pulling and umbrella sampling MD-simulations. After equilibration, position restraints were imposed on the $C_\alpha$ atoms of S6-helices (residues 553–583) to preserve the protein structure. Each THCV was pulled independently from the portal pockets along the directions that were defined as a line connecting the center of the pore gate (center of mass of $C_{\delta1}$ and $C_{\delta2}$ atoms of L574 of four protein subunits) and the center of a ligand (center of mass of heavy atoms of two central rings of THCV: $C_{04}$, $C_{05}$, $C_{06}$, $C_{07}$, $C_{08}$, $C_{12}$, $C_{13}$, $O_{02}$, $C_{14}$, see atom labels in Supplementary Fig. 3e, f). The pulling rate was set to 0.05 nm/ns and the force constant of 1000 kJ/(mol × nm²), because at these parameters the ligand moves along the protein surface and has enough mobility to adapt to the surface relief. As a result of 50 ns simulation run, we got MD trajectories describing dissociation of THCV, which were suitable for the potential of mean force (PMF) calculation with the umbrella sampling method. Then, 32 snapshots were taken from the pulling trajectory to generate starting configurations for umbrella sampling[83,84] windows along a collective variable (distance between the center of the pore gate and the ligand center). 5 ns MD-simulation with 1000 kJ/(mol × nm²) force constant constraining along the collective variable were performed for each window. The results were processed with the weighted histogram analysis method (WHAM)[85].

The approach of Molecular Hydrophobicity Potential (MHP) is a method that enables quantitative estimation of the spatial distribution of hydrophobic/hydrophilic properties, e.g., on a molecular surface. The formalism of MHP assumes that each atom in the molecular system is attributed with its specific hydrophobicity constant and the sum of atomic contributions is calculated at the molecular surface[52,86]. In this study, we used the atomic hydrophobicity constants obtained by Ghose et al.[86]. MHP calculations were performed with PLATINUM software[87].

## Fura-2 AM measurements

Full-length wild-type or mutant human TRPV6 was expressed in HEK 293S cells as described above. Forty-eight hours after transduction, cells were harvested by centrifugation at $550 \times g$ for 5 min. The cells were resuspended in prewarmed HEPES-buffered saline (HBS: 118 mM NaCl, 4.8 mM KCl, 1 mM $MgCl_2$, 5 mM D-glucose, 10 mM HEPES pH 7.4) containing 5 μg/ml of Fura-2 AM (Life Technologies) and incubated at 37 °C for 45 min. The loaded cells were then centrifuged for 5 min at $550 \times g$, resuspended again in prewarmed HBS, and incubated at 37 °C for 30 min in the dark. The cells were subsequently pelleted and washed twice, then resuspended in HBS for experiments. The cells were kept on ice in the dark for a maximum of ~2 h before fluorescence measurements, which were conducted using spectrofluorometer QuantaMaster 40 (Photon Technology International) at room temperature in a quartz cuvette under constant stirring. Intracellular $Ca^{2+}$

was measured by taking the ratio of fluorescence measurements at two excitation wavelengths (340 and 380 nm) and one emission wavelength (510 nm). The excitation wavelength was switched at 200-ms intervals. In experiments with THCV, every batch of HEK293 was split into two halves: half of cells were transfected with TRPV6 and the other half remained non-transfected. Measurements were repeated 4 to 8 times at each different concentration of THCV, which was added up to 100 μM concentration (from a 50 mM stock in 100% DMSO; higher than 100 μM concentrations of THCV were not used due to non-specific effects of THCV), and then the changes in fluorescence ($F_{340}/F_{380}$ ratios) recorded from non-transfected cells were subtracted from the changes in fluorescence recorded from transfected cells. For any single experiment with THCV, the antagonist was added to cells resuspended in HBS buffer 200 seconds prior to the addition of 10 mM calcium for activation in the recording (the last 50 seconds of the 200-s incubation were recorded as the baseline for each experimental recording trace). The normalized average of the corrected fluorescent signals was fitted with the logistic equation to determine the value of $IC_{50}$ (Fig. 2g). Data analysis was performed using Origin 9.1.0 (OriginLab Corp.).

### Electrophysiology

DNA encoding wild-type or mutant human TRPV6 was introduced into a plasmid for expression in eukaryotic cells that was engineered to produce GFP via a downstream internal ribosome entry site[88]. HEK 293 cells (ATCC #CRL-1573) grown on glass coverslips in 35-mm dishes were transiently transfected with 1-5 μg of plasmid DNA using Lipofectamine 2000 reagent (Invitrogen). Recordings were made at room temperature, 36 to 72 h after transfection. Currents from whole cells, typically held at a 0 or −60 mV membrane potential, were recorded using Axopatch 200B amplifier (Molecular Devices, LLC), filtered at 5 kHz, and digitized at 10 kHz using low-noise data acquisition system Digidata 1440 A and pCLAMP software (Molecular Devices, LLC). The external solution contained 140 mM NaCl, 6 mM CsCl, 1 mM $MgCl_2$, 10 mM HEPES pH 7.4 and 10 mM glucose. To evoke monovalent currents, 1 mM EGTA was added to the external solution. The pipette solution contained (in mM): 100 CsAsp, 20 CsF, 10 EGTA, 3 $MgCl_2$, 4 NaATP and 20 HEPES pH 7.2. TRPV6 currents were recorded in response to 50-ms voltage ramps from −100 to 70 mV. Data analysis was performed using Origin 9.1.0 (OriginLab Corp.).

### Fluorescence-detection size-exclusion chromatography

The experimental setup to conduct FSEC experiments in order to evaluate protein expression levels and tetrameric assembly of the channel was previously described in detail[59]. Full-length wild-type or mutant human TRPV6 was expressed in HEK cells as described above. Forty-eight hours after transduction, cells were harvested by centrifugation at 550 × g for 5 min. 1-ml aliquots of cells were solubilized under constant stirring for 1 h at 4 °C in 0.5 ml ice-cold lysis buffer containing 1% (w/v) n-dodecyl β-D-maltoside, 0.1% (w/v) CHS, 20 mM Tris-HCl pH 8.0, 150 mM NaCl, 0.8 μM aprotinin, 4.3 μM leupeptin, 2 μM pepstatin A, 1 mM phenylmethylsulfonyl fluoride, and 1 mM βME. The non-solubilized material was pelleted in a Sorvall MTX 150 Micro-Ultracentrifuge (Thermo Fisher Scientific) using an S100AT4 rotor for 30 min at 66,000 × g and 4 °C before 100 μl of the supernatant was injected into a Superose™ 6 10/300 GL SEC column mounted onto a Shimadzu HPLC system that included RF-10AXL fluorescence detector (Shimadzu). The SEC column was equilibrated with a buffer containing 1 mM n-dodecyl β-D-maltoside, 20 mM Tris-HCl pH 8.0, 150 mM NaCl, and 1 mM βME.

### Reporting summary

Further information on research design is available in the Nature Portfolio Reporting Summary linked to this article.

## Data availability

All data needed to evaluate the conclusions of the paper are present in the paper or the Supplementary Data. The cryo-EM density map of hTRPV6 in complex with THCV was deposited to the Electron Microscopy Data Bank (EMDB) under the accession code EMD-40676 (hTRPV6$_{THCV}$). The atomic coordinates have been deposited to the Protein Data Bank (PDB) under the accession code 8SP8 (hTRPV6$_{THCV}$; see Supplementary Table 1). The accession codes for previously published structures that were used for model building, MD simulations, and/or illustrations: 7S88, 7S89, 7S8B, 7S8C, 6U88, and 7UGG. PDB THCV coordinate (THCV_conf2.mol2) and topology (THCV.itp and extra.itp) files used in MD simulations and coordinates of the protein and ligands obtained in MD simulations (see also Supplementary Table 2) are available as Supplementary Data 1 and Supplementary Data 2. All other data are available from the corresponding author upon request. Source data are provided with this paper.

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

## Acknowledgements

We thank Sean Mulligan (Pacific Northwest Center for Cryo-EM) for help with microscope operation and data collection. A portion of this research was supported by NIH grant U24GM129547 and performed at the PNCC at OHSU and accessed through EMSL (grid.436923.9), a DOE Office of Science User Facility sponsored by the Office of Biological and Environmental Research. A.N. is a Walter Benjamin Fellow funded by the Deutsche Forschungsgemeinschaft (DFG, German Research Foundation)–464295817. MD simulations were supported by the RSF (23-14-00313). A.I.S. was supported by the NIH (R01 AR078814, R01 CA206573, R01 NS083660, R01 NS107253) and NSF (1818086).

## Author contributions

A.N. carried out protein expression, protein purification, and cryo-EM data processing. A.N., J.K. and L.S.K made constructs. Y.A.T., N.A.K. and R.G.E. designed computational work, performed molecular modeling, and analyzed the data. M.V.Y. carried out electrophysiological experiments. A.N. and K.D.N. prepared cryo-EM samples. A.N., K.D.N. and L.S.K. carried out Fura-2 intracellular calcium imaging. J.K. performed FSEC experiments. A.N. and A.I.S. analyzed structural data and built molecular models. A.N., Y.A.T. and A.I.S. wrote the manuscript, which was then edited by all authors.

## Competing interests

The authors declare no competing interests.
