## [Peer Review File · Nature Communications]

Molecular pathway and structural mechanism of human oncochannel TRPV6 inhibition by the phytocannabinoid tetrahydrocannabivarinREVIEWER COMMENTS

Reviewer #1 (Remarks to the Author):

In this manuscript, Neuberger et al. present a cryo-EM structure of the calcium channel TRPV6 in the presence of the cannabinoid antagonist tetrahydrocannabivarin (THCV). Since the structure suggests two possible THCV interaction sites, the authors further performed limited mutagenesis and a complex set of MD simulations, from which they propose mechanisms of ligand binding and channel inhibition. The manuscript contains high-quality data and the topic is of interest. However, the novelty is somewhat limited, since structures of (1) the same channel with several other antagonists (2-APB, ruthenium red, econazole, PCHPDs), or (2) of related TRPV channels with cannabinoids are already available from earlier work of this team and others. Moreover, as outlined below, the mutagenesis and MD experiments have significant limitations, reducing the strength of the conclusions that can be drawn.

Specific points:

1) Figure 1b,c, and lines 97-100:

Here, the authors report that the inhibition of calcium influx TRPV6 by THCV is incomplete, even at very high doses. This seems in contrast with patch-clamp data in this paper and in ref. 35, as well as with calcium imaging data in ref. 35 (although species differences may be contributing). What is missing to make this claim solid are important controls: (1) responses to calcium addition in the presence of THCV in non- or mock-transfected cells, and (2) responses to calcium addition in the presence of a full and potent pore blocker (e.g. cadmium).

2) Probing of the THCV binding site – lines 145 and further:

Here, "To verify the THCV binding site", the authors made mutations to three leucines (L490, L574, L568), "which are in close proximity to the observed THCV binding location". Two of these leucines are mutated to tryptophan, and one to tyrosine.

There is no real rationale provided why these three residues were chosen, or why two were mutated to tryptophan and one to tyrosine. As indicated further in the manuscript, L574 is not even pointing towards the binding site, but towards the pore, so the verification of the proposed binding site is actually based on only two mutations.

If the authors want to provide strong evidence for the exact binding site and the critical residues, several further experiments/data are needed, such as: (1) some quantitative data to support the choice of these residues; (2) more systematic mutations to other (potentially) critical residues in the proposed binding pocket (e.g. based on Extended data Fig. 2), as well as mutations of the same residues to other amino acids (e.g. conservative mutations); (3) a more detailed functional analysis of these mutants.

3) Following up on this last point: the functional data for the mutants shown in Figure 2f, which are central to the further experiments and discussion in the paper, raise many questions (see also my point 1). First, is the inhibition of L568W by THCv significant at any of the tested doses? In my opinion, the data do not seem robust enough to make the conclusion that the concentration dependency is shifted to lower concentrations compared to WT. Second, the slope of the concentration-inhibition curve for L490W is much shallower than for the other mutants. What could be an explanation for that? It also seems that whereas WT was tested at concentrations up to 300 μM , L490W and L568W were not tested (or shown?) higher than 30 μM . If the authors want to compare maximal inhibition that can be obtained for these channels, the same maximal concentration should be used. Third, mutagenesis may also affect antagonist potency or efficacy by an allosteric effect. In that regard, it would be important to show how these mutants affect the antagonism of other ligands that bind at different sites in the channel, along with a more detail description of their overall gating properties.

4) MD simulations:

As a non-specialist in MD simulations, I found this part interesting but very difficult to follow. For instance, while this may be clear for a specialist, I wonder how parameters such as the pulling rate or force constant were chosen, and how the choice would affect the final outcomes. In any case, the obtained results would benefit from at least some sort of experimental validation. For instance, the proposed horizontal pathway could be challenged with selected mutations and/or close structural analogues of THCv.

5) Lines 238-240 : “The significantly diminished ability of THCv to inhibit calcium entry through the L568W mutant is likely due to limited access of the ligand to the portal site via bulky tryptophan side chain obstruction of the horizontal route.”

I don't think this statement makes much sense. If the access to the binding site is difficult, that would result in a lower affinity (reduced k_{on}) or at least a slower inhibition of the channel (if k_{on} and k_{off} are both equally affected), but not on the maximal inhibition that can be obtained.

6) Lines 245-248 : “The side chain of Y574, however, can block the pore more effectively than the side chain of leucine when the bulkier tyrosine is in across-the-pore orientation (Fig. 4j). Correspondingly, THCv binding to a single portal binding site in the L574Y mutant can cause the increased efficiency of THCv inhibition observed experimentally (Fig. 2f).”

What is meant here? Do the authors mean to suggest that binding of a single THCv molecule would be sufficient to block the mutant channel, whereas binding of more THCv molecules is needed to block the WT channel? Please clarify. In any case, it would be important to evaluate whether this channel is then also more sensitive to other antagonists.

7) THCv inhibition of TRPV5, a close relative of TRPV6 (75% overall amino acid sequence identity), was shown to slow the progression of joint destruction in a rat model for osteoarthritis, highlighting a new potential in therapeutic application of THCv for treatment of osteoarthritis⁴⁶.

Ref. 46 does not seem to be a reasonable reference for this statement. Please revise.

Reviewer #2 (Remarks to the Author):

This study by Neuberger and colleagues describes interaction between TRPV6 cation channel and plant cannabinoid called tetrahydrocannabinol (THCv) by functional and structural analyses, which include patch clamp, ratiometric Ca imaging, cryo-EM and MD simulation. Cannabinoid analogs are known ligands of the thermo-TRPs (V1-V4, M8 and A1, see PMID: 30697147) and have been studied for treatment of psycho- or neuro-related disorders. On this regard, their relation to TRPV6 is little explored despite of recently reported inhibition of TRPV6 channel function by THCv. This manuscript investigates structural interaction of THCv with TRPV6 using cryo-EM and MD simulations to correlate with the inhibitory effect of THCv and compare with the known complexing of two other antagonists (ruthenium red and econazole) with TRPV6. High resolution structure of TRPV6/THCv obtained by cryo-EM allows identification of a closed state in which hydrophobic gate residue M578 defines a much narrower pore constriction in comparison with the apo (or open) state. The authors also identify two THCv binding sites within the tetrameric channel protein and residues that interact/bind with THCv. Together with MD simulations, mutagenesis and functional analysis, the authors conclude that site 1 as the primary THCv binding site is located between the central pore cavity and the membrane surrounding the channel protein. I found that the presented data are solid and in general are adequately interpreted. Thus, this study provides novel and insightful knowledge on interaction between a cannabinoid ligand and TRPV6. I have some specific comments, as outlined below.

1. Based on functional effects of tryptophan mutation at L490, L574 and L568 of the THCv binding site 1 the authors conclude that the data “strongly support site 1 as a primary site for TRPV6 inhibition by THCv” (Line 156). It would be more careful to draw the conclusion if the function of some other mutant(s) is determined as control/comparison, eg, tryptophan mutation to some residue(s) of the binding site #2.

2. TRPV6 does have known agonists or activators such as 11-hydroxy-THCv (THCv-OH) and PIP₂. Thus, although the apo state of TRPV6 is an open state, in the presence of 11-hydroxy-THCv or PIP₂, the channel would be in an activated state which, I wonder, would correspond to an increased open probability (open more often) or increased single-channel conductance (larger pore size, ie, open wider than the apo state) or both. Because of structural similarity between THCv and THCv-OH, it'd be helpful to add discussion on how THCv-OH would change the pore conformation and the conformation of the lower part of S6 (still α helix?) to correlate with functional increase/activation. Further, MD simulations using THCv-ON, if realistic for this paper, would provide insightful data as well.

Minor comments.

1. While TRPV6 is known to be a Ca selective cation channel (as also mentioned in the manuscript), it's in fact also permeable to monovalent cations including Na. This justifies the experiments for Fig. 1a data using monovalent cation-containing solutions. But this information should be indicated in the figure legend (although it's described in Methods) because people may naturally think that Ca is used as the main permeant.

2. Lines 178-180, forward and backward orientations. Orientations should be clearly indicated/labeled in the figure panel or legend. It's unclear to me which figure panel(s) shows THCV orientations.

3. Line 468, "internal solution", is this pipette or intracellular solution? If the intracellular solution is supposed to be the same as the pipette solution, please add a description on how to achieve it.

4. Line 541, panel b, numbers in x-axis are misplaced.

Reviewer #3 (Remarks to the Author):

The manuscript "Molecular pathway and structural mechanism of human onco-channel TRPV6 inhibition by the natural phytocannabinoid tetrahydrocannabivarin" by Neuberger et al reports the structure of human TRPV6 in complex with the cannabinoid tetrahydrocannabivarin (THCV), and complementing functional data as well as molecular dynamics simulations. As such, there is no doubt this study is important to stimulate development of new TRPV6-targeting drugs, and valuable for understanding the function of cannabinoid-based ligands on TRPV channels in general. However, I have several concerns regarding the manuscript. A key aspect is the fact that the authors themselves refer the THCV binding site as putative (and I agree with them), despite overall relatively high resolution of the cryo-EM data. The authors have tried to overcome this issue with validating mutational analysis through assessment of the intracellular calcium levels, but of three selected residues, one is already known to line the ion conducting pore, and hence effects on the flux must be anticipated, which may not necessarily relate to THCV binding. These issues are problematic since the entire following manuscript assumes the assigned THCV binding site is correct. I also find the binding site poorly described, and several of the described statements regarding the structure lack supporting figures. I hope the below-mentioned comments can be used to further strengthen the manuscript.

Major issues:

The main topic of the manuscript is about inhibition of TRPV6, yet there is little support in the introduction to show the value of inhibition of over-expressed TRPV6, while instead numerous examples of dysfunctional TRPV6 are outlined. This seems somewhat out of focus.

Regarding the putative binding sites, I am lacking more detailed close-views than currently shown in panel 2c. I suggest to also include all interacting residues in the figures, including supporting cryo-EM at the signal-to-noise (SNR) level already shown, and also with higher a SNR to appreciate if the features are more or less well-supported than the surrounding residues. This will nicely complement Extended Fig. 2. Please also provide half-maps densities of the binding sites.

The binding sites of several TRPV6 inhibitors or activators have been reported and two are mentioned in the discussion. It would be relevant with a structural comparison of the THCv site with other ligands of TRPV6 and perhaps also with cannabinoids binding to other TRP channels. This would be valuable for understanding if modulatory effects are maintained between compounds.

For functional characterization, I advise to include an assessment of the protein production levels using e.g. Western-blot, as differences between protein forms may well otherwise obscure the analysis. Similarly, I recommend employing an alternative inhibitor for validation of the functionality. It is problematic that L574 was targeted as a mutation site to validate site 1 considering it lines the ion conductance pore and hence can be expected to be sensitive for mutagenesis (despite its THCv concentration dependency). Related, I wonder if the selection of L568 as a mutation site may also be questioned, as it is part of the THCv induced π to alpha transition of helix S6. In this light, I suggest assessing two or preferably three more residues the line the binding site to validate the site, considering that the cryo-EM data are inconclusive. In Figure 2c, L573 appears closer to THCv than L574, and hence L573 represent one residue to be targeted. I recommend using conservative changes that ensure that the chemical environment is preserved, while at the same time reducing THCv binding capacity.

Associated, the authors used a truncated TRPV6 version for the structural characterization to avoid issues related to CaM binding. Did this truncation affect the TRPV6 function? Is it possible that THCv can bind to this region? Can the full-length protein be included in the functional analysis as a control?

From “Comparing the open-state hTRPV6 structure to the structure solved in the presence of THCv ...” and until “Formation of this hydrogen bond with THCv in the backward orientation leads to dissociation of the ligand from the binding pocket.” several statements in the text require supportive figures, or it is very difficult for the reader to assess the validity. This also holds true even if the remarks are consistent with previous studies.

In the last Results section a THCv inhibition mechanism is proposed based on comparisons in-between TRPV6THCv and TRPV6open. Here it would be suitable to include complementary views and to correlate

this with the effects on the pore and Fig. 3 (other than the views from the intracellular side), including supporting cryo-EM density. How does the inhibition mechanism compare to other inhibitors of TRPV6?

Minor issues:

- 1) The signal-to-noise (σ) level of the cryo-EM density should be indicated throughout.

- 2) Panel d is missing in Fig. 2. Fig. 2e seems to refer to a single point mutation, but the text suggests all mutations were assessed?

- 3) The relevance of Fig. 4 is not clear. Fig. 4c lacks the residues indicated in the legend.

- 4) A data processing procedure figure is missing. An Euler distribution plots is missing.

Reviewer #4 (Remarks to the Author):

In their study "Molecular pathway and structural mechanism of human oncochannel TRPV6 inhibition by the natural phytocannabinoid tetrahydrocannabivarin", Neuberger and colleagues identify a binding site for the TRPV6 inhibitor THCv, a naturally occurring cannabinoid. The calcium-selective TRPV6 plays an important role in calcium homeostasis and may be a key target for antagonist treatment of various cancers. This study should address a broad audience as there is great interest in TRP channels as well as cannabinoids for therapeutic use.

Major concerns:

The authors used mutagenesis to validate the binding site for THCv identified by cryo-EM and supported by MD simulation studies. However, a more thorough functional characterization is needed as mutagenesis may not only affect the THCv binding site but also affect the channel integrity including changed channel conducting properties.

1. Current-voltage curves (voltage ramps) should be performed for the mutants (L490W, L568W and L574Y) and compared to wt. This would indicate any obvious change in channel properties (peak current and kinetics), assuming the expression levels of membrane incorporated channels do not differ for the various constructs. Otherwise, single-channel recording could resolve this issue. Please also provide information regarding cytoplasmic membrane expression levels for wt and mutants.

2. Concentration-response curves for calcium should be performed for the mutants (L490W, L568W and L574Y) and compared to wt. Even though similar control fluorescence ratios are achieved for wt and L574Y (Figs 1 and 2), it could be that wt is more sensitive than L574Y. Thus, 10 mM may be a too high calcium concentration for wt (supramaximal stimulation) explaining the different IC50 values obtained for wt and L574Y, and why wt is also not completely inhibited by THCV. What about fluorescence ratios for the other constructs? This information is important not only for adjusting the calcium response to avoid supramaximal stimulation, but also as an indication of any changed channel conducting properties (EC50 and Emax) as a complement to current measurements as discussed above. This important information is lacking with normalized values. Furthermore, in case of overstimulated wt TRPV6, maybe THCV could still completely inhibit wt under these conditions unless its solubility limits higher concentrations to be tested? What is the solubility of THCV? Most likely poor $\geq 100 \mu\text{M}$. Please also provide information on THCV vehicle (DMSO?) and its effects in electrophysiology and calcium-imaging studies.

3. What about the effect of other small-molecule TRPV6 inhibitors such as 2-APB, econazole, PCHPDs and ruthenium red on the mutants? Could 2-APB and econazole be used to shed light on possible unwanted effects caused by the mutations used to define the binding site for THCV? Likewise, ruthenium red and PCHPDs could be used to confirm that complete inhibition of wt and mutants (L490W and L568W) can indeed be achieved?

Minor concerns:

Line 28: TRPV6 is an important but not the major driver?

Line 274-276: "Compared to CBD and dyclonine, THCV appears to have a unique among TRPV channels allosteric mechanism of inhibition." The reasoning is unclear. Have the authors tried CBD? Maybe it binds to the same site and exerts allosteric inhibition similar to THCV?

In the title, "natural" is not needed as phytocannabinoid is a naturally occurring cannabinoid.

Please specify the incubation time for THCV in functional and cryo-EM studies.

Dose-response should be concentration-response (dose is used for intact organism).

Reviewer #5 (Remarks to the Author):

The manuscript by Neuberger et al. reports on an interesting investigation on the mechanism of inhibition of TRPV6 by tetrahydrocannabivarin. In particular, a combination of cryoEM, electrophysiology and MD simulations is used to characterize the binding of tetrahydrocannabivarin to the wild type and to selected mutants. The outcomes of this investigation are potentially of significance, since the molecular mechanism of modulation of TRPV channels by ligands and drugs is still partially unclear. While I find the paper to be overall well written and the approach to be rigorous and well described, I think that the work still misses a clear connection between ligand binding and stabilization of the closed state. Therefore I find that the conclusions of the paper (emphasized by the title "Molecular pathway and structural mechanism of ... inhibition...") are not fully supported by the evidence provided by the authors.

In particular, while a thorough characterization of the binding pathway and binding mode of tetrahydrocannabivarin is provided on the basis of MD simulations, the allosteric mechanism responsible for inhibition is not at all addressed. The same simulations could have been done on the open configuration to test whether or not opening of the channel leads to a destabilization of the bound state at site 1 using site 2 as a reference. Furthermore per-residues estimates of the interaction energy between the drug and the channel in the two states should be provided to test the hypothesis that the drug "pulls" on few selected side chains from S6 causing the alpha-to-pi transition. Overall, I do not think that the manuscript is ready for publication and additional work should be done to clarify the allosteric mechanism.

We are very thankful to Reviewers for their excellent suggestions. We have made changes in the manuscript accordingly with the details outlined in our responses below.

Reviewer #1 (Remarks to the Author)

In this manuscript, Neuberger et al. present a cryo-EM structure of the calcium channel TRPV6 in the presence of the cannabinoid antagonist tetrahydrocannabivarin (THCV). Since the structure suggests two possible THCV interaction sites, the authors further performed limited mutagenesis and a complex set of MD simulations, from which they propose mechanisms of ligand binding and channel inhibition. The manuscript contains high-quality data and the topic is of interest. However, the novelty is somewhat limited, since structures of (1) the same channel with several other antagonists (2-APB, ruthenium red, econazole, PCHPDs), or (2) of related TRPV channels with cannabinoids are already available from earlier work of this team and others. Moreover, as outlined below, the mutagenesis and MD experiments have significant limitations, reducing the strength of the conclusions that can be drawn.

We thank Reviewer #1 for kind words about the quality of the data. With respect to novelty: (1) While TRPV6 structures are indeed available with other antagonists, all previously studied antagonists were shown to bind to sites different from THCV (see new **Supplementary Fig. 7a,b**). The THCV site in TRPV6 is therefore novel, not reported before. (2) The cannabinoid cannabidiol (CBD) has been shown to bind to a similar site in TRPV2, but functions as an agonist, not antagonist. The finding that the portal site in TRPV6 is an antagonist (not agonist!) binding site is novel.

Specific points:

1) Figure 1b,c, and lines 97-100:

Here, the authors report that the inhibition of calcium influx TRPV6 by THCV is incomplete, even at very high doses. This seems in contrast with patch-clamp data in this paper and in ref. 35, as well as with calcium imaging data in ref. 35 (although species differences may be contributing). What is missing to make this claim solid are important controls: (1) responses to calcium addition in the presence of THCV in non- or mock-transfected cells, and (2) responses to calcium addition in the presence of a full and potent pore blocker (e.g. cadmium).

As suggested by Reviewer #1, we performed experiments by measuring calcium influx in response to different concentrations of THCV using HEK293 cells from the same batch, half of them transfected with TRPV6 (see example in panel **a** in the figure below) and half non-transfected (panel **b**). For both transfected and non-transfected cells, we made 4 to 8 measurements at each concentration of THCV and then subtracted the signals for non-transfected cells from the signals for transfected cells and plotted the average (panel **c**). The fitting of such corrected fluorescent signals with logistic equation (curve through the points in panel **c**) gave THCV concentration dependencies that predict complete inhibition for the wild type and the majority of mutant TRPV6 channels (see updated **Figure 2g**). The corresponding procedure has now been described in the Methods section (lines 501-512).

We performed these experiments not only for 7 new mutants such as L568Q but have also repeated them for the wild type and 3 previously reported mutants (L490W, L568W and L574Y). Importantly, while for the majority of tested mutants we now see the complete inhibition by THCv, our qualitative conclusions have not changed from the original submission. For example, the new IC_{50} for WT is $15.3 \pm 2.3 \mu\text{M}$ compared to $14.6 \pm 3.1 \mu\text{M}$ reported in the original version of the manuscript. The L568W mutant, for which we previously reported very small inhibition by THCv, in fact shows no TRPV6 function at all, because non-specific changes in fluorescence caused by THCv (like in panel **b** in the figure above) completely cancelled out changes observed in the transfected cells (see panel **a** in the figure below). Similarly, one of our newly made mutants, C494W, also turned out to be non-functional (see panel **b** in the figure below). We also ran whole-cell patch-clamp experiments and neither L568W nor C494W produced any measurable currents, strongly supporting the lack of function demonstrated in the calcium uptake experiments.

Previously we performed similar calcium uptake experiments with the TRPV6 blocker Gd^{3+} (doi:10.1038/nature17975) and showed the full block of the signal (higher affinity of Gd^{3+} allows to use lower concentrations of this blocker that do not cause non-specific effects). Since the conditions of the experiment were the same, we decided to not repeat the experiments with Gd^{3+} .

2) Probing of the THCv binding site – lines 145 and further:

Here, "To verify the THCv binding site", the authors made mutations to three leucines (L490, L574, L568), "which are in close proximity to the observed THCv binding location". Two of these leucines are mutated to tryptophan, and one to tyrosine.

There is no real rationale provided why these three residues were chosen, or why two were mutated to tryptophan and one to tyrosine. As indicated further in the manuscript, L574 is not even pointing

towards the binding site, but towards the pore, so the verification of the proposed binding site is actually based on only two mutations.

We absolutely agree with Reviewer #1 that mutating L574 to verify THCv binding was a wrong idea. We have therefore removed all the results of experiments conducted with L574Y from the manuscript.

We have now made mutations of nearly all residues (nine mutations in total, **Fig. 2g**) predicted to contribute to THCv binding (**Supplementary Figure 3e,f**). Most residues were mutated to alanine, while alanine A561 and leucines L490 and L568 were mutated to the bulky tryptophan. We used fluorescence-detection size-exclusion chromatography (FSEC) to confirm expression and tetrameric assembly of the wild type and mutant channels produced in HEK 293 cells (**Supplementary Figure 4a-h**) and tested their function using Fura-2 AM-based measurements of intracellular calcium (**Fig. 2e-g**). Two mutants, C494W and L568W, did not show measurable calcium uptake (see figure above). Confirming that these mutants were not functional, whole-cell recordings from HEK 293 cells expressing these mutants did not detect measurable TRPV6-mediated currents either. For example, mutants that did show calcium uptake also demonstrated typical TRPV6-mediated currents in electrophysiological recordings (**Supplementary Figure 4i-l**). As an alternative to the L568W mutation, we substituted leucine L568 with glutamine and the resulting L568Q mutant turned out to be functional.

Our calcium uptake measurement showed that all mutants can be divided into two major groups: those that demonstrated a rightward shift in the THCv concentration dependence or weakening of the inhibition (behavior exemplified by A561W in the left panel below) and those that showed a leftward shift in the concentration dependence or increased potency of THCv (behavior exemplified by I564A in the middle panel below). We have 3 mutants with the rightward shift and 4 with the leftward shift (see the right panel below).

Overall, the changes in the concentration dependence of calcium uptake inhibition by THCv observed in mutant compared to wild-type channels strongly support THCv binding to sites 1 and 2. Thus, replacement of the large hydrophobic side chain of M497 with the short side chain of alanine as well as substitution of hydrophobic leucine L568 with hydrophilic glutamine resulted in reduced potency of THCv, likely due to the loss of hydrophobic interactions. On the other hand, weakening of THCv inhibition observed for the A561W mutant most probably originated from more restricted access to the portals imposed by the bulky side chain of tryptophan compared to the small side chain of alanine. On the opposite, smaller side chains of alanine in F493A, I564A and T567A likely made it easier for the THCv molecule to reach the deep portal site 1 or have created more room and opportunity for the inhibitor to form optimal hydrophobic interactions with the surrounding protein, thus leading to an increased potency of the inhibitor (leftward shift in the THCv concentration dependence). Interestingly, replacement of leucine L490 that points towards sites 1 and 2 with the bulky tryptophan produces a dual effect: it increases THCv potency, likely due to increased number of hydrophobic interactions with

the indole ring in one rotameric conformation, and at the same time it makes the inhibition ~50% incomplete, probably because the indole ring in an alternative rotameric conformation fills up the portal space and prevents binding of THCv. In this regard, a small ~20% fraction of uninhibited F493A channels at high THCv concentration may originate from an alternative positioning of the THCv molecule in the roomier site 1, where it does not cause conformational changes anymore that lead to channel closure. The corresponding information has been added to the manuscript as updated panels **e-g** in **Figure 2** as well as changes in the text (lines 153-194).

If the authors want to provide strong evidence for the exact binding site and the critical residues, several further experiments/data are needed, such as: (1) some quantitative data to support the choice of these residues; (2) more systematic mutations to other (potentially) critical residues in the proposed binding pocket (e.g. based on Extended data Fig. 2), as well as mutations of the same residues to other amino acids (e.g. conservative mutations); (3) a more detailed functional analysis of these mutants.

We have now mutated most of the residues contributing to sites 1 and 2. Of the total 9 mutants, 2 turned out to be non-functional, while the majority of the remaining 7 showed altered THCv potency (**Fig. 2g**): 4 demonstrated a leftward shift in THCv concentration dependence, while 3 displayed a rightward shift. All the observed effects are consistent with THCv binding to the identified sites, as described above. The residue L568 was mutated to two different residues, tryptophan and glutamine; L568W was non-functional (see above), while L568Q showed reduced THCv potency (**Fig. 2g**). We also performed detailed functional analysis of the mutants. Apart from using calcium uptake measurements to characterize THCv inhibition (**Fig. 2e-g**), we recorded current-voltage relationships for select mutants using whole-cell patch-clamp recordings (**Supplementary Fig. 4i-l**) as well as measured calcium concentration dependencies of TRPV6 activation (**Supplementary Fig. 4m**) and concentration dependencies of inhibition by econazole (**Supplementary Fig. 4n**). The extent of current inhibition by THCv observed in electrophysiological experiments (**Supplementary Fig. 4i-l**) was roughly proportional to the THCv inhibition of calcium uptake (**Fig. 2g**). For the mutants that showed strongest changes in THCv potency, the EC₅₀ values for calcium (**Supplementary Fig. 4m**) and IC₅₀ values for econazole (**Supplementary Fig. 4n**) did not change dramatically, suggesting that the majority of functional properties of these mutant channels were not drastically affected by the corresponding mutations.

3) Following up on this last point: the functional data for the mutants shown in Figure 2f, which are central to the further experiments and discussion in the paper, raise many questions (see also my point 1). First, is the inhibition of L568W by THCv significant at any of the tested doses? In my opinion, the data do not seem robust enough to make the conclusion that the concentration dependency is shifted to lower concentrations compared to WT.

As mentioned above (see specific point 1), after subtracting the non-specific changes in fluorescence for non-transfected cells caused by THCv, it turned out that the L568W mutant is non-functional (also confirmed by electrophysiological recordings). Correspondingly, data for L568W was completely removed from the figure and the fact that this mutant (along with C494W) is non-functional has been mentioned in the text (lines 159-164).

Second, the slope of the concentration-inhibition curve for L490W is much shallower than for the other mutants. What could be an explanation for that?

After subtracting the non-specific changes in fluorescence for non-transfected cells caused by THCV, the slope of the concentration-inhibition curve for L490W is about the same as for other mutant and wild type channels (see **Fig. 2g**).

It also seems that whereas WT was tested at concentrations up to 300 μM , L490W and L568W were not tested (or shown?) higher than 30 μM . If the authors want to compare maximal inhibition that can be obtained for these channels, the same maximal concentration should be used.

We have now used exactly the same set of concentrations for all mutant and wild type channels (1 – 100 μM). Using concentrations higher than 100 μM is problematic for two reasons. First, there is a significant increase in non-specific fluorescence for non-transfected cells. Second, the high price of the THCV reagent is prohibitive for us to run so many experiments.

Third, mutagenesis may also affect antagonist potency or efficacy by an allosteric effect. In that regard, it would be important to show how these mutants affect the antagonism of other ligands that bind at different sites in the channel, along with a more detail description of their overall gating properties.

For mutants that showed strongest changes in THCV potency (**Fig. 2g**), we measured calcium concentration dependencies of TRPV6 activation (**Supplementary Fig. 4m**) and concentration dependencies of inhibition by econazole (**Supplementary Fig. 4n**). The EC_{50} values for calcium (**Supplementary Fig. 4m**) and IC_{50} values for econazole (**Supplementary Fig. 4n**) did not change dramatically, suggesting that the majority of functional properties of these mutant channels were not drastically affected by the corresponding mutations. The corresponding information has been added to the text (lines 187-191).

4) MD simulations:

As a non-specialist in MD simulations, I found this part interesting but very difficult to follow. For instance, while this may be clear for a specialist, I wonder how parameters such as the pulling rate or force constant were chosen, and how the choice would affect the final outcomes. In any case, the obtained results would benefit from at least some sort of experimental validation. For instance, the proposed horizontal pathway could be challenged with selected mutations and/or close structural analogues of THCV.

The choice of parameters for pulling simulations (the so-called steered MD) is still an empirical approach based on literature data and our own experience in relation to similar systems. In this work, we performed a series of preliminary simulations with a pulling rate varying in the range of 0.01-10 nm/ns, and the same constant force equal to 1000 kJ/(mol \times nm²). If the pulling rate is too high, the ligand breaks away from the protein surface and leaves the site following a straight line. If it is too slow, the ligand trajectory becomes very “bumpy”, which complicates the choice of initial states for umbrella sampling simulations. The force constant of 1000 kJ/(mol \times nm²) is typical for this kind of simulation (for example, see Lemkul *et al.*, 2010 (reference [81])). We used the velocity of 0.05 nm/ns and the constant force of 1000 kJ/(mol \times nm²); at these parameter values, the ligand moved along the protein surface and had sufficient mobility to (1) adapt to the relief and (2) form a smooth trajectory. As a result, we obtained two trajectories for THCV dissociation that are suitable for calculating the potential of mean force (PMF) by the umbrella sampling method. Given the largely empirical nature of this approach, we would like to note that the comparative analysis of alternative trajectories describing the dissociation of the ligand from the site along different paths is important here. This is exactly what has been done in this manuscript. We have added a short clarification on this in the Methods section (see lines 472-487).

5) Lines 238-240 :“The significantly diminished ability of THCv to inhibit calcium entry through the L568W mutant is likely due to limited access of the ligand to the portal site via bulky tryptophan side chain obstruction of the horizontal route.”

I don't think this statement makes much sense. If the access to the binding site is difficult, that would result in a lower affinity (reduced k_{on}) or at least a slower inhibition of the channel (if k_{on} and k_{off} are both equally affected), but not on the maximal inhibition that can be obtained.

Since after signal correction, the L568W mutant turned out to be non-functional, the corresponding data and text have been removed from the manuscript.

6) Lines 245-248 : “The side chain of Y574, however, can block the pore more effectively than the side chain of leucine when the bulkier tyrosine is in across-the-pore orientation (Fig. 4j). Correspondingly, THCv binding to a single portal binding site in the L574Y mutant can cause the increased efficiency of THCv inhibition observed experimentally (Fig. 2f).”

What is meant here? Do the authors mean to suggest that binding of a single THCv molecule would be sufficient to block the mutant channel, whereas binding of more THCv molecules is needed to block the WT channel? Please clarify. In any case, it would be important to evaluate whether this channel is then also more sensitive to other antagonists.

Since L574 does not contribute to THCv binding all the results of experiments with L574Y have been removed from the manuscript.

7) THCv inhibition of TRPV5, a close relative of TRPV6 (75% overall amino acid sequence identity), was shown to slow the progression of joint destruction in a rat model for osteoarthritis, highlighting a new potential in therapeutic application of THCv for treatment of osteoarthritis⁴⁶.

Ref. 46 does not seem to be a reasonable reference for this statement. Please revise.

Thank you for pointing out the error in referencing. We have now fixed this mistake and cite the right paper (ref. 58: <https://pubmed.ncbi.nlm.nih.gov/28535500/>).

Reviewer #2 (Remarks to the Author)

This study by Neuberger and colleagues describes interaction between TRPV6 cation channel and plant cannabinoid called tetrahydrocannabinol (THCv) by functional and structural analyses, which include patch clamp, ratiometric Ca imaging, cryo-EM and MD simulation. Cannabinoid analogs are known ligands of the thermo-TRPs (V1-V4, M8 and A1, see PMID: 30697147) and have been studied for treatment of psycho- or neuro-related disorders. On this regard, their relation to TRPV6 is little explored despite of recently reported inhibition of TRPV6 channel function by THCv. This manuscript investigates structural interaction of THCv with TRPV6 using cryo-EM and MD simulations to correlate with the inhibitory effect of THCv and compare with the known complexing of two other antagonists (ruthenium red and econazole) with TRPV6. High resolution structure of TRPV6/THCv obtained by cryo-EM allows identification of a closed state in which hydrophobic gate residue M578 defines a much narrower pore constriction in comparison with the apo (or open) state. The authors also identify two THCv binding sites within the tetrameric channel protein and residues that interact/bind with THCv. Together with MD simulations, mutagenesis and functional analysis, the authors conclude that site 1 as the primary THCv

binding site is located between the central pore cavity and the membrane surrounding the channel protein. I found that the presented data are solid and in general are adequately interpreted. Thus, this study provides novel and insightful knowledge on interaction between a cannabinoid ligand and TRPV6. I have some specific comments, as outlined below.

We thank Reviewer #2 for the kind assessment of our work.

1. Based on functional effects of tryptophan mutation at L490, L574 and L568 of the THCv binding site 1 the authors conclude that the data “strongly support site 1 as a primary site for TRPV6 inhibition by THCv” (Line 156). It would be more careful to draw the conclusion if the function of some other mutant(s) is determined as control/comparison, eg, tryptophan mutation to some residue(s) of the binding site #2.

We have now mutated most of the residues contributing to sites 1 and 2. Of the total 9 mutants, 2 turned out to be non-functional, while the majority of the remaining 7 showed altered THCv potency (Fig. 2g): 4 demonstrated a leftward shift in THCv concentration dependence, while 3 displayed a rightward shift. All the observed effects are consistent with THCv binding to the identified sites, as described above.

2. TRPV6 does have known agonists or activators such as 11-hydroxy-THCv (THCv-OH) and PIP2. Thus, although the apo state of TRPV6 is an open state, in the presence of 11-hydroxy-THCv or PIP2, the channel would be in an activated state which, I wonder, would correspond to an increased open probability (open more often) or increased single-channel conductance (larger pore size, ie, open wider than the apo state) or both. Because of structural similarity between THCv and THCv-OH, it'd be helpful to add discussion on how THCv-OH would change the pore conformation and the conformation of the lower part of S6 (still π helix?) to correlate with functional increase/activation. Further, MD simulations using THCv-OH, if realistic for this paper, would provide insightful data as well.

We are grateful to the Reviewer for this question. It is indeed very important to test our model of ligand binding using a molecule with similar structure but different activity. In this case, THCv-OH is an excellent candidate for such an assessment. To address this issue, we carried out MD-simulations with THCv-OH in the binding pockets using the same protocol as for THCv. Our simulations suggest that the two hydroxyl groups of THCv-OH tend to form hydrogen bonds with L490 and T567 (see figure below). Due to the symmetrical arrangement of hydroxyl groups, the displacement of the center of mass of THCv-OH molecule in forward and backward orientations is minimal and similar to the displacement of THCv in the forward orientation. Correspondingly, based on our MD simulations, we hypothesize that THCv-OH can bind to site 1 at least as well as THCv. We think that the ligand activates TRPV6 by some mechanism not directly related to the inhibition via site 1, and this mechanism blocks the putative inhibition of THCv-OH as a side effect.

Regarding the biophysical mechanism of PIP₂ and THCV-OH, these molecules are likely to increase P_o and not the single-channel conductance. Our attempts to visualize PIP₂ bound to TRPV6 structurally have not been successful so far.

Minor comments.

1. While TRPV6 is known to be a Ca selective cation channel (as also mentioned in the manuscript), it's in fact also permeable to monovalent cations including Na. This justifies the experiments for Fig. 1a data using monovalent cation-containing solutions. But this information should be indicated in the figure legend (although it's described in Methods) because people may naturally think that Ca is used as the main permeant.

As suggested, we have added the corresponding information to the figure legend (lines 577-578).

2. Lines 178-180, forward and backward orientations. Orientations should be clearly indicated/labeled in the figure panel or legend. It's unclear to me which figure panel(s) shows THCV orientations.

We are grateful to Reviewer #2 for the suggestion. The orientations have now been labeled in **Figure 4** panels **a** and **b** and indicated in the figure legend (lines 625-627).

3. Line 468, "internal solution", is this pipette or intracellular solution? If the intracellular solution is supposed to be the same as the pipette solution, please add a description on how to achieve it.

Internal solution is the pipette solution, which in the whole-cell mode becomes the intracellular solution due to diffusion. The corresponding correction has been made in the text (line 517-519).

4. Line 541, panel b, numbers in x-axis are misplaced.

Thank you for noticing. The labels have been fixed.

Reviewer #3 (Remarks to the Author)

The manuscript "Molecular pathway and structural mechanism of human onco-channel TRPV6 inhibition by the natural phytocannabinoid tetrahydrocannabivarin" by Neuberger et al reports the structure of human TRPV6 in complex with the cannabinoid tetrahydrocannabivarin (THCV), and complementing functional data as well as molecular dynamics simulations. As such, there is no doubt this study is

important to stimulate development of new TRPV6-targeting drugs, and valuable for understanding the function of cannabinoid-based ligands on TRPV channels in general.

We thank Reviewer #3 for the kind assessment of our work.

However, I have several concerns regarding the manuscript. A key aspect is the fact that the authors themselves refer the THCv binding site as putative (and I agree with them), despite overall relatively high resolution of the cryo-EM data. The authors have tried to overcome this issue with validating mutational analysis through assessment of the intracellular calcium levels, but of three selected residues, one is already known to line the ion conducting pore, and hence effects on the flux must be anticipated, which may not necessarily relate to THCv binding. These issues are problematic since the entire following manuscript assumes the assigned THCv binding site is correct. I also find the binding site poorly described, and several of the described statements regarding the structure lack supporting figures. I hope the below-mentioned comments can be used to further strengthen the manuscript.

Since L574 does not contribute to THCv binding, all the results of experiments conducted with L574Y have been removed from the manuscript. Instead, we have mutated most of the residues contributing to sites 1 and 2. Of the total 9 mutants, 2 turned out to be non-functional, while the majority of the remaining 7 showed altered THCv potency (**Fig. 2g**): 4 demonstrated a leftward shift in THCv concentration dependence, while 3 displayed a rightward shift. All the observed effects are consistent with THCv binding to the identified sites, as described above and in the manuscript. To better describe the THCv binding site, we replaced the former panel **c** in **Fig. 2** with two closeup orthogonal views (new panels **c** and **d** in **Fig. 2**) that show the binding region in more detail.

Major issues:

The main topic of the manuscript is about inhibition of TRPV6, yet there is little support in the introduction to show the value of inhibition of over-expressed TRPV6, while instead numerous examples of dysfunctional TRPV6 are outlined. This seems somewhat out of focus.

Thank you for this comment. We have revised the Introduction accordingly (lines 65-74).

Regarding the putative binding sites, I am lacking more detailed close-views than currently shown in panel 2c. I suggest to also include all interacting residues in the figures, including supporting cryo-EM at the signal-to-noise (SNR) level already shown, and also with higher a SNR to appreciate if the features are more or less well-supported than the surrounding residues. This will nicely complement Extended Fig. 2. Please also provide half-maps densities of the binding sites.

We have replaced the former panel **c** in **Fig. 2** with two closeup orthogonal views (new panels **c** and **d** in **Fig. 2**) that show the binding region in more detail. We have also added maps at lower and higher SNR as well as half maps to **Supplementary Figure 3**.

The binding sites of several TRPV6 inhibitors or activators have been reported and two are mentioned in the discussion. It would be relevant with a structural comparison of the THCv site with other ligands of TRPV6 and perhaps also with cannabinoids binding to other TRP channels. This would be valuable for understanding if modulatory effects are maintained between compounds.

To facilitate the requested comparison of ligand-binding sites, we have made a new supplementary figure (**Supplementary Fig. 7**) that shows THCv binding site in TRPV6 simultaneously with other binding

sites in TRPV6 (a-b) or illustrates ligands at the portal site of TRPV2 (cannabidiol, c-d) and TRPV3 (dyclonine, e-f).

For functional characterization, I advise to include an assessment of the protein production levels using e.g. Western-blot, as differences between protein forms may well otherwise obscure the analysis.

As an assessment of the protein production level, we have now presented fluorescence-detection size-exclusion chromatography (FSEC) data (new **Supplementary Fig. 4a-h**), where the height of the tetrameric peak reflects the amount of properly folded TRPV6 tetramer. As clearly seen from **Supplementary Fig. 4a-h**, the expression of the properly folded tetrameric form of TRPV6 is similar in all mutant and wild type channels included in our study.

Similarly, I recommend employing an alternative inhibitor for validation of the functionality.

We selected mutants that showed the strongest changes in THCv potency (**Fig. 2g**) and tested their inhibition by an alternative TRPV6 inhibitor, econazole. As clearly shown in **Supplementary Figure 4n**, the IC_{50} for the concentration dependence of the mutant channels was similar to the value for wild type channels. The corresponding information has been added to the text (lines 187-194, 680-686).

It is problematic that L574 was targeted as a mutation site to validate site 1 considering it lines the ion conductance pore and hence can be expected to be sensitive for mutagenesis (despite its THCv concentration dependency). Related, I wonder if the selection of L568 as a mutation site may also be questioned, as it is part of the THCv induced π to alpha transition of helix S6. In this light, I suggest assessing two or preferably three more residues the line the binding site to validate the site, considering that the cryo-EM data are inconclusive. In Figure 2c, L573 appears closer to THCv than L574, and hence L573 represent one residue to be targeted. I recommend using conservative changes that ensure that the chemical environment is preserved, while at the same time reducing THCv binding capacity.

Since L574 does not contribute to THCv binding, all the results of experiments conducted with L574Y have been removed from the manuscript. Instead, we have mutated most of the residues contributing to sites 1 and 2. Of the total 9 mutants, 2 turned out to be non-functional, while the majority of the remaining 7 showed altered THCv potency (**Fig. 2g**): 4 demonstrated a leftward shift in THCv concentration dependence, while 3 displayed a rightward shift. All the observed effects are consistent with THCv binding to the identified sites, as described above and in the manuscript (lines 153-194).

Associated, the authors used a truncated TRPV6 version for the structural characterization to avoid issues related to CaM binding. Did this truncation affect the TRPV6 function? Is it possible that THCv can bind to this region? Can the full-length protein be included in the functional analysis as a control?

All functional experiments in this manuscript have been done with the full-length protein, not truncated. The truncated version was only used for structural experiments. According to the previous work (refs. 30, 31, 39, 40), the only function that has been shown different between the full-length and truncated TRPV6 is CaM-dependent inactivation, while other functions are similar. The corresponding citation of the previous work has been added to the text of the manuscript (line 108-110).

From “Comparing the open-state hTRPV6 structure to the structure solved in the presence of THCv ...” and until “Formation of this hydrogen bond with THCv in the backward orientation leads to dissociation of the ligand from the binding pocket.” several statements in the text require supportive figures, or it is

very difficult for the reader to assess the validity. This also holds true even if the remarks are consistent with previous studies.

We added clarifications in the text of the MD section (lines 216-286) and additional details in **Fig. 4**.

In the last Results section a THCV inhibition mechanism is proposed based on comparisons in-between TRPV6THCV and TRPV6open. Here it would be suitable to include complementary views and to correlate this with the effects on the pore and Fig. 3 (other than the views from the intracellular side), including supporting cryo-EM density. How does the inhibition mechanism compare to other inhibitors of TRPV6?

Complementary views have been included into **Figure 5** (panels **b** and **c**). Cryo-EM density is now illustrated in **Figure 1d-f** and **Supplementary Figures 2e** and **3a-d**. The mechanisms of TRPV6 inhibition by THCV, 2-APB, econazole and RR are similar in terms of the resulting conformation of the inhibited state (see **Figure 3b**). However, since binding sites for TRPV6 inhibitors are different (**Supplementary Figure 7a-b**), the allosteric mechanisms that drive the channel closure are likely different as well. The hypothetical mechanism of THCV inhibition is illustrated in **Figure 5**.

Minor issues:

1) The signal-to-noise (sigma) level of the cryo-EM density should be indicated throughout.

The sigma level has been indicated in **Supplementary Figure 3a-d**.

2) Panel d is missing in Fig. 2. Fig. 2e seems to refer to a single point mutation, but the text suggests all mutations were assessed?

We have fixed this typo.

3) The relevance of Fig. 4 is not clear. Fig. 4c lacks the residues indicated in the legend.

The relevance of **Figure 4** has been described in more detail in the text (lines 216-286). The residues mentioned in the text have now been shown and labeled in panels **a** and **b** of the updated **Figure 4**.

4) A data processing procedure figure is missing. An Euler distribution plots is missing.

The data processing procedure has now been described in the new **Supplementary Figure 1**. The Euler distribution has been added to the new **Supplementary Figure 2** as panel **d**.

Reviewer #4 (Remarks to the Author)

In their study "Molecular pathway and structural mechanism of human oncochannel TRPV6 inhibition by the natural phytocannabinoid tetrahydrocannabivarin", Neuberger and colleagues identify a binding site for the TRPV6 inhibitor THCV, a naturally occurring cannabinoid. The calcium-selective TRPV6 plays an important role in calcium homeostasis and may be a key target for antagonist treatment of various cancers. This study should address a broad audience as there is great interest in TRP channels as well as cannabinoids for therapeutic use.

We thank Reviewer #4 for emphasizing the importance of our study.

Major concerns:

The authors used mutagenesis to validate the binding site for THCv identified by cryo-EM and supported by MD simulation studies. However, a more thorough functional characterization is needed as mutagenesis may not only affect the THCv binding site but also affect the channel integrity including changed channel conducting properties.

1. Current-voltage curves (voltage ramps) should be performed for the mutants (L490W, L568W and L574Y) and compared to wt. This would indicate any obvious change in channel properties (peak current and kinetics), assuming the expression levels of membrane incorporated channels do not differ for the various constructs. Otherwise, single-channel recording could resolve this issue. Please also provide information regarding cytoplasmic membrane expression levels for wt and mutants.

We have now provided current-voltage curves for wild type and select mutant channels in the new **Supplementary Figure 4i-l**. As an assessment of the cytoplasmic membrane expression level, we have presented fluorescence-detection size-exclusion chromatography (FSEC) data for wild type and mutant channels in the new **Supplementary Figure 4a-h**.

2. Concentration-response curves for calcium should be performed for the mutants (L490W, L568W and L574Y) and compared to wt. Even though similar control fluorescence ratios are achieved for wt and L574Y (Figs 1 and 2), it could be that wt is more sensitive than L574Y. Thus, 10 mM may be a too high calcium concentration for wt (supramaximal stimulation) explaining the different IC₅₀ values obtained for wt and L574Y, and why wt is also not completely inhibited by THCv. What about fluorescence ratios for the other constructs? This information is important not only for adjusting the calcium response to avoid supramaximal stimulation, but also as an indication of any changed channel conducting properties (EC₅₀ and E_{max}) as a complement to current measurements as discussed above. This important information is lacking with normalized values. Furthermore, in case of overstimulated wt TRPV6, maybe THCv could still completely inhibit wt under these conditions unless its solubility limits higher concentrations to be tested? What is the solubility of THCv? Most likely poor $\geq 100 \mu\text{M}$. Please also provide information on THCv vehicle (DMSO?) and its effects in electrophysiology and calcium-imaging studies.

We selected mutants that showed the strongest changes in THCv potency (**Fig. 2g**) and measured Ca²⁺ concentration dependence for them. As clearly shown in **Supplementary Figure 4m**, the EC₅₀ for the concentration dependence of the mutant channels was similar to wild type channels. The corresponding information has been added to the text (lines 187-191, 680-686). The issue of incomplete block has been resolved by subtracting non-specific signals measured as the fluorescence changes recorded from non-transfected cells in response to applications of THCv. The stock of THCv was prepared in DMSO, in which the compound was soluble 50 mM concentration. The final concentration of DMSO in our THCv-containing buffers did not exceed 100 μM . This concentration of DMSO did not affect our current or fluorescence measurements in electrophysiological or Fura experiments. The corresponding information has been added to the Methods section (lines 394-396, 501-510).

3. What about the effect of other small-molecule TRPV6 inhibitors such as 2-APB, econazole, PCHPDs and ruthenium red on the mutants? Could 2-APB and econazole be used to shed light on possible unwanted effects caused by the mutations used to define the binding site for THCv? Likewise, ruthenium red and PCHPDs could be used to confirm that complete inhibition of wt and mutants

(L490W and L568W) can indeed be achieved?

We characterized inhibition of three mutants, which showed the strongest effects on THCv potency, by econazole and found complete inhibition and no significant change in econazole potency compared to wild type channels (**Supplementary Fig. 4n**). L568W mutant was not functional (see responses to Reviewer #1 above). After correction for THCv-induced changes in fluorescence, the THCv concentration dependencies predicted complete inhibition for the wild type and the majority of mutant TRPV6 channels (see **Figure 2g**).

Minor concerns:

Line 28: TRPV6 is an important but not the major driver?

We agree with the Reviewer and corrected the text accordingly (line 29).

Line 274-276: "Compared to CBD and dyclonine, THCv appears to have a unique among TRPV channels allosteric mechanism of inhibition." The reasoning is unclear. Have the authors tried CBD? Maybe it binds to the same site and exerts allosteric inhibition similar to THCv?

We implied novelty compared to the already published structural mechanisms. We have corrected the text to avoid confusion (lines 312-314). Lack of CBD effect on TRPV6 was already shown before (see Janssens *et al.* 2018 [40]).

In the title, "natural" is not needed as phytocannabinoid is a naturally occurring cannabinoid.

"Natural" has been removed from the title.

Please specify the incubation time for THCv in functional and cryo-EM studies.

The incubation time has now been mentioned for both functional and cryo-EM experiments (lines 394-396, 507-510).

Dose-response should be concentration-response (dose is used for intact organism).

Agreed. The corresponding correction has been made in the text (lines 583 and 602).

Reviewer #5 (Remarks to the Author):

The manuscript by Neuberger et al. reports on an interesting investigation on the mechanism of inhibition of TRPV6 by tetrahydrocannabivarin. In particular, a combination of cryoEM, electrophysiology and MD simulations is used to characterize the binding of tetrahydrocannabivarin to the wild type and to selected mutants. The outcomes of this investigation are potentially of significance, since the molecular mechanism of modulation of TRPV channels by ligands and drugs is still partially unclear. While I find the paper to be overall well written and the approach to be rigorous and well described, I think that the work still misses a clear connection between ligand binding and stabilization of the closed state. Therefore, I find that the conclusions of the paper (emphasized by the title "Molecular pathway and structural mechanism of ... inhibition...") are not fully supported by the

evidence provided by the authors.

In particular, while a thorough characterization of the binding pathway and binding mode of tetrahydrocannabivarin is provided on the basis of MD simulations, the allosteric mechanism responsible for inhibition is not at all addressed. The same simulations could have been done on the open configuration to test whether or not opening of the channel leads to a destabilization of the bound state at site 1 using site 2 as a reference. Furthermore per-residues estimates of the interaction energy between the drug and the channel in the two states should be provided to test the hypothesis that the drug "pulls" on few selected side chains from S6 causing the alpha-to-pi transition. Overall, I do not think that the manuscript is ready for publication and additional work should be done to clarify the allosteric mechanism.

We thank Reviewer #5 for the valuable comment. Allosteric effects can indeed play an important role in the regulation of channels through ligands binding. According to the Reviewer's suggestion, we conducted an additional MD simulation with THCV embedded in sites 1 of apo (open) TRPV6 in an orientation similar to the cryo-EM model. As a result, we show that the ligands left the pockets immediately after the removal of the positional restrictions imposed on them. We therefore came to the conclusion that the location of the ligand in the site of the open TRPV6 is unstable. Moreover, three of the four replicas of the ligand came out of the pockets along the so-called "horizontal" route proposed in this study. This result once again confirms the horizontal route as the most likely path for the ligand to approach or leave its portal binding pocket (we have described this result in the text, lines 216-286, and **Supplementary Fig. 5**). We also calculated the average interaction energies (Coulomb + LJ) between THCV at site 1 and the protein in the inhibited and open states (the energies were calculated as the average of four THCV molecules on the second 10 ns trajectories when the positions of the ligands were fixed; see Methods). We have shown that the interaction energy is significantly higher for the open-state TRPV6 ($\Delta E = +9$ kJ/mol), which is thus in excellent agreement with the poor stability of the ligand when modeling free MD. We believe that the main contribution to the energy difference is determined by the change in the orientation of the carbonyl oxygen atom L490: in the open channel, it forms an intra-helix hydrogen bond, thus weakening the interaction with the hydroxyl group of THCV, while in TRPV6_{THCV}, the carbonyl oxygen rotates towards the site due to the kink in the helix S5, which strengthens the THCV-L490 hydrogen bond (see **Fig. 4a** and **Supplementary Fig. 5a**). We understand that the estimates made for ΔE are based on the values of the force field energy values, and not on the free energy. A more or less correct assessment of the latter can be made if the entropy factor is taken into account. This is a non-trivial task for such a large (four protein subunits in a hydrated lipid bilayer) and dynamic (with many functional states) system. This task requires special consideration and therefore goes beyond the scope of this study. The same is true for the computational evaluation of allosteric effects. Moreover, there are still no reliable methods for this – the area is under active research development. To avoid confusion with the free energy estimates, we have not included the values of ΔE in the text. In any case, as mentioned above, even such a simple parameter supports the trend obtained with non-restrained MD, which predicts fast escape of the THCV molecule from site 1 in the open TRPV6 channel.

REVIEWER COMMENTS

Reviewer #1 (Remarks to the Author):

The authors have made clear improvements to the manuscript, primarily by adding new mutagenesis data and refining the calcium imaging assay, and I reiterate that the structural data are of high quality. Nevertheless, I still have reservations regarding the novelty and broad general importance of this work to warrant publication in Nature Communications instead of a journal more focused on detailed structural analyses.

1) While I fully acknowledge that a structure of TRPV6 with THCv is novel, this remains somewhat incremental given that (1) structures of TRPV6 with a plethora of other ligands have already been published, (2) that the inhibition of TRPV6 by THCv and other ligands has already been published, and (3) that binding sites for cannabinoids on related TRPV channels have already been published. One also has to acknowledge that the affinity of the channel for THCv is relatively low ($IC_{50} \sim 14$ micromols), that THCv has more potent activity towards a multitude of other targets, and that the cryoEM structure is not of sufficient resolution to determine the exact orientation of THCv in the binding site(s) of the channel. Therefore, extrapolating these findings to the development of new TRPV6-targeting drugs is a very long stretch.

2) I still consider the functional analysis of the mutant channels to be quite superficial. For instance, in Supplementary Figure 4i-l, the authors show whole-cell patch-clamp recordings of WT and mutant TRPV6. It is clear from these examples that the mutations have a dramatic effect on the whole cell current properties. For instance, the three mutants (in contrast to WT) exhibit quite significant outward current, and mutants M497A and L490W largely lose the typical channel rectification. As the mechanisms of TRPV5/6 rectification have been studied in great detail in older studies, where it was shown that these are properties of the selectivity filter, one wonders how these mutations, far away from the selectivity filter, can have such effects.

3) I also still have doubts regarding the suitability of the calcium assay to test alterations in THCv potency and maximal inhibitory efficacy. The authors now state that "changes in fluorescence recorded from non-transfected cells were subtracted from the changes in fluorescence recorded from transfected cells." It is unclear what they mean by this: did the authors subtract the individual F340 and F380 signals in control cells from the individual F340 and F380 signals in transfected cells, or did they subtract ratios from ratios? Please specify. In both cases, this does not seem ideal for a ratiometric dye such as Fura2, since the relation between ratio and calcium is highly non-linear. Very often, cells overexpressing TRPV6 or other TRP channels have already elevated basal calcium levels, and any change in basal calcium level would affect the delta-ratio signal, even if the actual signal is not altered. These assays also lack a positive control to demonstrate that full inhibition can be achieved. This seems especially important for mutant L490W, where the authors claim incomplete inhibition yet higher affinity, and where the patch-

clamp measurements seem to indicate strongly altered gating properties (see my point 2). At least for this mutant, a full characterization in patch-clamp and demonstration of full inhibition using a pore blocker such as Gd3+ would be essential to support the authors' model.

Reviewer #2 (Remarks to the Author):

The authors made additional point mutants related the two THCv binding sites and performed functional analyses. The data and interpretations have been improved and they no longer used 'PRIMARY THCv binding site' to describe the site 1. MD simulation on THCv-OH has been performed in comparison with THCv.

The revised manuscript has much been improved.

Reviewer #3 (Remarks to the Author):

The authors have adequately responded to the issues raised in my first evaluation of the manuscript and the paper has improved significantly – congratulations! In particular, the authors have mutated additional residues that line the THCv binding sites, and assessed the expression levels, thereby validating the sites detected using cryo-EM. Moreover, additional figures have been introduced to visualize the sites and the supporting cryo-EM density. In this light, the manuscript is suitable for publication in my opinion.

Minor remaining comment: I disagree with the authors that L490W is expressed as the other TRPV6 forms. Assuming similar levels of sample were injected for the FSEC analysis (as shown in Supplementary Fig. 4 and described in the methods), it rather looks as if 5x lower levels are present which may well affect the functional analysis and interpretation of this particular mutant. Has this been accounted for in the functional assay?

Reviewer #4 (Remarks to the Author):

In their rebuttal letter, the authors of this interesting study have addressed all minor concerns and some of the major concerns regarding possible global changes of channel property due to mutagenesis. Whereas the initial key mutants L568W and L574Y are non-functional and concluded not relevant for THCv binding, respectively, the L490W mutant is still of great interest. Indeed, the formation of a hydrogen bond between L490 and THCv seems critical for the inhibitory effect of this cannabinoid. It is therefore surprising that L490W was not characterized in more detail including calcium-dependent activation and the inhibitory effect of econazole or pore inhibitors. The dual effect of L490W on THCv (increased potency and 50% reduced I_{max}) could well be as proposed by the authors, also favoring that THCv binding to site 1 inhibits TRPV6 (lines 273-286). However, additional functional experiments are needed to consolidate a key and selective role of L490 in THCv inhibition.

Major concerns:

Calcium concentration-dependent responses (F340/F380 traces and curves) and econazole concentration-response curves must be performed for L490W, as done for some other mutants that showed the strongest changes in THCv potency (Suppl. Fig .4).

Please define what “strongest changes” mean; change in IC_{50} and/or I_{max} . For example, THCv caused a larger leftward shift in F493A compared to I564A, in which however THCv caused the most pronounced maximal inhibition (I_{max}). Likewise, THCv may be more potent in L490W than I564A according to IC_{50} values.

Thus, please provide statistics to support mutant differences compared to WT (Fig. 2). For example, is T567A really different from WT “...made it easier for the THCv molecule to reach the deep portal site 1...” (lines 174-178). More importantly, is there a true effect of L490W (IC_{50} and at each concentration of THCv tested) compared to WT (Fig. 2g)? As stated, voltage ramps for L490W and some other mutants displayed typical TRPV6-mediated currents (line 162). However, L490W, M497A and L568Q but not WT, displayed strong outward currents at positive voltages. The shape of the L490W current differed the most from WT, and surprisingly there was no inhibition of L490W currents by THCv (25 μ M), which caused equal inhibition of calcium-activated responses in L490W and WT (Fig. 2g). This should be discussed. Although, the inhibitory effect of THCv (25 μ M) in M497A and L568Q as well as WT seems to correlate between voltage ramp and calcium measurements, these mutants also displayed changed behavior.

It is unfortunate that basal calcium signals (non-transfected cells) in the studies of L490W are much higher than in studies of the other mutants (Source data file), perhaps further complicating the analysis of L490 in THCv binding?

Other concerns:

In the revised manuscript, fluorescence data (control) are presented after subtraction of the non-specific changes in fluorescence for non-transfected cells caused by THCv (lines 506-507). Is this also the case for experiments in Suppl. Fig. 4m,n? No controls are presented in Source data file.

The slope of the concentration-response curve to econazole is different in M497A compared to other mutants (Suppl. Fig. 4n). What could be the reason?

Concentration of calcium is 10 mM (Suppl Fig. 4n)? Add to legend.

For electrophysiology and calcium measurements; define n and independent experiments (cell batches, transfections, replicates).

Error bars indicate? Please add "data are represented as the mean \pm SEM/SD" or similar to legends (Figs 1 and 2, Suppl. Fig. 4). Software for calculation of mean, SD/SEM, IC50?

Voltage ramps: -120 to +120 mV (Fig. 1 legend, results and methods) or -100 to +70 mV (Fig. 1a and Suppl. Fig. 4i-l)? Number of experiments/cells (Suppl. Fig. 4i-l)? Can IC50 values be calculated as in Fig. 1a? $H_p = 0$ or -60 mV (see methods).

The same data for calcium measurements are used in Fig. 1c and Fig. 2g. This should be stated within legends.

Suppl. Fig. 4n; Source data file does not support n = 12 for WT, rather n = 9.

Arrow direction should be opposite in rotational views (Fig. 1d-f and Fig. 2a-d)?

Reviewer #5 (Remarks to the Author):

I am satisfied with the revision of the manuscript. The additional simulations address properly the concerns that I raised and provide further support to the conclusions. I now recommend publication of the manuscript.

We thank the Reviewers for their additional comments and suggestions aimed at further strengthening our study. We have made changes in the manuscript accordingly, with details outlined in our responses below.

Reviewer #1 (Remarks to the Author)

The authors have made clear improvements to the manuscript, primarily by adding new mutagenesis data and refining the calcium imaging assay, and I reiterate that the structural data are of high quality. Nevertheless, I still have reservations regarding the novelty and broad general importance of this work to warrant publication in Nature Communications instead of a journal more focused on detailed structural analyses.

1) While I fully acknowledge that a structure of TRPV6 with THCv is novel, this remains somewhat incremental given that (1) structures of TRPV6 with a plethora of other ligands have already been published, (2) that the inhibition of TRPV6 by THCv and other ligands has already been published, and (3) that binding sites for cannabinoids on related TRPV channels have already been published. One also has to acknowledge that the affinity of the channel for THCv is relatively low ($IC_{50} \sim 14$ micromols), that THCv has more potent activity towards a multitude of other targets, and that the cryoEM structure is not of sufficient resolution to determine the exact orientation of THCv in the binding site(s) of the channel. Therefore, extrapolating these findings to the development of new TRPV6-targeting drugs is a very long stretch.

Again, we respectfully disagree with Reviewer #1 on the novelty of THCv site: (1) while a few binding sites have been reported in TRPV6, the sites of THCv have never been identified in TRPV6, (2) while it was indeed shown that THCv inhibits TRPV6, it was not known how and through which site, and (3) to our knowledge, binding sites for only two cannabinoids, cannabidiol (CBD) and phytocannabinoid Δ^9 -tetrahydrocannabinol (THC), were reported, both acting as activators of a relatively far-related TRPV channel rat TRPV2. In particular, since previous studies have already demonstrated that CBD (as well as a number of other cannabinoids) do not modulate TRPV6, there is virtually nothing that can be derived from the study on TRPV2's activation by cannabinoids. To summarize, in our case, we have a different channel (TRPV6), a different cannabinoid (THCv), and an inhibitory effect, which is opposite to activation. Given the importance of cannabinoids as perspective medications, it is our firm belief that the novelty of THCv's binding to and allosteric inhibition of human TRPV6 is sufficiently high.

2) I still consider the functional analysis of the mutant channels to be quite superficial. For instance, in Supplementary Figure 4i-l, the authors show whole-cell patch-clamp recordings of WT and mutant TRPV6. It is clear from these examples that the mutations have a dramatic effect on the whole cell current properties. For instance, the three mutants (in contrast to WT) exhibit quite significant outward current, and mutants M497A and L490W largely lose the typical channel rectification. As the mechanisms of TRPV5/6 rectification have been studied in great detail in older studies, where it was shown that these are properties of the selectivity filter, one wonders how these mutations, far away from the selectivity filter, can have such effects.

Compared to a lot of purely structural studies that are being published, we have invested a significant amount of time and effort to provide multidisciplinary analyses of THCv interaction with human TRPV6, namely combining state-of-the-art molecular dynamic simulation with a variety of functional methods. Moreover, as requested by several reviewers in the first round of revision, we have introduced and tested almost every residue in proximity to THCv bound to the portal site of TRPV6. Beyond testing the effects of these mutations on THCv's modulation of TRPV6, we have also conducted other experiments

to study these mutants (FSEC, calcium uptake measurements from untransfected cells for baseline activity correction, calcium and econazole concentration dependencies, and now Gd^{3+} inhibition). With all this work done, we admit that some of the previous examples of current-voltage relationships for the mutant channels were recorded at $n = 1$ and for two of them, L490W and M497A, were not typical (most likely due to large leak currents). We repeated these experiments so that we have more than 3 independent repeats for every mutant, and now provide more typical examples for L490W and M497A in Supplementary Fig. 4j–k. Based on all our recordings, we conclude that the current-voltage relationships for mutant channels are grossly similar to wild-type IV curves. Of course, detailed studies of the voltage-dependencies may reveal fine differences between wild type and some of the mutants, but we consider these details to be outside the scope of the present study.

3) I also still have doubts regarding the suitability of the calcium assay to test alterations in THCV potency and maximal inhibitory efficacy. The authors now state that "changes in fluorescence recorded from non-transfected cells were subtracted from the changes in fluorescence recorded from transfected cells." It is unclear what they mean by this: did the authors subtract the individual F340 and F380 signals in control cells from the individual F340 and F380 signals in transfected cells, or did they subtract ratios from ratios? Please specify. In both cases, this does not seem ideal for a ratiometric dye such as Fura2, since the relation between ratio and calcium is highly non-linear. Very often, cells overexpressing TRPV6 or other TRP channels have already elevated basal calcium levels, and any change in basal calcium level would affect the delta-ratio signal, even if the actual signal is not altered. These assays also lack a positive control to demonstrate that full inhibition can be achieved. This seems especially important for mutant L490W, where the authors claim incomplete inhibition yet higher affinity, and where the patch-clamp measurements seem to indicate strongly altered gating properties (see my point 2). At least for this mutant, a full characterization in patch-clamp and demonstration of full inhibition using a pore blocker such as Gd^{3+} would be essential to support the authors' model.

We are confident that after subtraction of signal using the same but non-transfected cell batch, we completely eliminated non-specific signal for THCV effect on non-transfected cells. The subtraction was done for the F340 to F380 ratios. The corresponding explanation has been added to the text (line 526). As requested by the Reviewer, we have also carried out additional experiments with L490W. (1) We measured concentration dependence of TRPV6 activation by Ca^{2+} and showed that the EC_{50} value for L490W (1.33 ± 0.04 mM) is similar to the value for wild-type TRPV6 (1.05 ± 0.03 mM). (2) As requested by the Reviewer, we have also measured the concentration dependence of TRPV6 inhibition by econazole and showed that the IC_{50} value for L490W (10.7 ± 0.7 μ M) is about twice larger than the value for wild-type channel (4.7 ± 0.1 μ M). This slight weakening of econazole inhibition is likely due to allosteric effect of the L490W mutation on econazole binding. Indeed, L490 is located on the opposite site of the S5 helix, in which the W495 side chain makes direct contact with the econazole molecule. (3) Finally, we have performed calcium uptake measurements in the presence of Gd^{3+} and show that application of 100 mM Gd^{3+} results in complete inhibition of calcium uptake through L490W channels ($n = 3$). These data add yet more experimental evidence to our previously performed experiments and show once more that this mutant forms a tetrameric channel functionally similar to wild type. The new data has been added to panels m–o of the Supplementary Figure 4. The corresponding information has been added to the text (lines 202–214, 706–719).

Reviewer #2 (Remarks to the Author)

The authors made additional point mutants related the two THCv binding sites and performed functional analyses. The data and interpretations have been improved and they no longer used 'PRIMARY THCv binding site' to describe the site 1. MD simulation on THCv-OH has been performed in comparison with THCv.

The revised manuscript has much been improved.

We thank Reviewer #2 for the kind assessment of our work.

Reviewer #3 (Remarks to the Author)

The authors have adequately responded to the issues raised in my first evaluation of the manuscript and the paper has improved significantly – congratulations! In particular, the authors have mutated additional residues that line the THCv bindings sites, and assessed the expression levels, thereby validating the sites detected using cryo-EM. Moreover, additional figures have been introduced to visualize the sites and the supporting cryo-EM density. In this light, the manuscript is suitable for publication in my opinion.

We thank Reviewer #3 for the kind words about our manuscript.

Minor remaining comment: I disagree with the authors that L490W is expressed as the other TRPV6 forms. Assuming similar levels of sample were injected for the FSEC analysis (as shown in Supplementary Fig. 4 and described in the methods), it rather looks as if 5x lower levels are present which may well affect the functional analysis and interpretation of this particular mutant. Has this been accounted for in the functional assay?

We have now mentioned the lower expression level of L490W (lines 161–163). This indeed leads to somewhat lower fluorescent signals for this mutant. Nevertheless, the Fura-2 AM fluorescent signals for L490W were strong enough to make reliable measurements of intracellular calcium. Confirming this, the error values for L490W were comparable to wild type and other mutants (Fig. 2g, Supplementary Fig. 4m-n). We therefore believe that the lower expression level of this mutant did not compromise our results.

Reviewer #4 (Remarks to the Author)

In their rebuttal letter, the authors of this interesting study have addressed all minor concerns and some of the major concerns regarding possible global changes of channel property due to mutagenesis. Whereas the initial key mutants L568W and L574Y are non-functional and concluded not relevant for THCv binding, respectively, the L490W mutant is still of great interest. Indeed, the formation of a hydrogen bond between L490 and THCv seems critical for the inhibitory effect of this cannabinoid. It is therefore surprising that L490W was not characterized in more detail including calcium-dependent activation and the inhibitory effect of econazole or pore inhibitors. The dual effect of L490W on THCv (increased potency and 50% reduced I_{max}) could well be as proposed by the authors, also favoring that THCv binding to site 1 inhibits TRPV6 (lines 273-286). However, additional functional experiments are needed to consolidate a key and selective role of L490 in THCv inhibition.

As suggested, we have carried out additional experiments with L490W, measured concentration dependence of TRPV6 activation by Ca^{2+} and showed that the EC_{50} value for L490W (1.33 ± 0.04 mM) is similar to the value for wild-type TRPV6 (1.05 ± 0.03 mM). We have also measured the concentration dependence of TRPV6 inhibition by econazole and showed that the IC_{50} value for L490W (10.7 ± 0.7 μ M) is about twice larger than the value for wild-type channel (4.7 ± 0.1 μ M). This slight weakening of econazole inhibition is likely due to allosteric effect of the L490W mutation on econazole binding. Indeed, L490 is located on the opposite site of the S5 helix, in which the W495 side chain makes direct contact with the econazole molecule. Finally, we have performed calcium uptake measurements in the presence of Gd^{3+} and show that application of 100 mM Gd^{3+} results in complete inhibition of calcium uptake through L490W channels ($n = 3$). The new data has been added to panels m-o of the Supplementary Figure 4. The corresponding information has been added to the text (lines 202–214, 706–719).

Major concerns:

Calcium concentration-dependent responses (F_{340}/F_{380} traces and curves) and econazole concentration-response curves must be performed for L490W, as done for some other mutants that showed the strongest changes in THCv potency (Suppl. Fig. 4).

Done. See above and new Supplementary Figure 4m–n.

Please define what “strongest changes” mean; change in IC_{50} and/or I_{max} . For example, THCv caused a larger leftward shift in F493A compared to I564A, in which however THCv caused the most pronounced maximal inhibition (I_{max}). Likewise, THCv may be more potent in L490W than I564A according to IC_{50}

values.

For these experiments, we selected two mutants with pronounced leftward (L490W and I564A) and rightward (M497A and A561W) shifts in IC_{50} for THC. The corresponding explanation has been added to the text (lines 202–207).

Thus, please provide statistics to support mutant differences compared to WT (Fig. 2). For example, is T567A really different from WT “...made it easier for the THC molecule to reach the deep portal site 1...” (lines 174-178). More importantly, is there a true effect of L490W (IC_{50} and at each concentration of THC tested) compared to WT (Fig. 2g)? As stated, voltage ramps for L490W and some other mutants displayed typical TRPV6-mediated currents (line 162). However, L490W, M497A and L568Q but not WT, displayed strong outward currents at positive voltages. The shape of the L490W current differed the most from WT, and surprisingly there was no inhibition of L490W currents by THC (25 μ M), which caused equal inhibition of calcium-activated responses in L490W and WT (Fig. 2g). This should be discussed. Although, the inhibitory effect of THC (25 μ M) in M497A and L568Q as well as WT seems to correlate between voltage ramp and calcium measurements, these mutants also displayed changed behavior.

We have provided the requested statistics by comparing the $\Delta(F_{340}/F_{380})$ values for wild type TRPV6 and L490W at 3 and 100 μ M THC and at 30 μ M THC for other mutants (lines 171-172, 178-180, 181-182, 188-189, 196–197). We found that the differences between T567A and wild type were not significant. Accordingly, we corrected the statement mentioned by Reviewer #4 (lines 184-189). Other concentrations dependencies showed significant differences compared to wild type. We have to admit that some of the previous examples of current-voltage relationships for the mutant channels were recorded at $n = 1$ and for two of them, L490W and M497A, were not typical (most likely due to large leak currents). We repeated these experiments and now provide more typical examples for L490W and M497A in Supplementary Fig. 4j–k. Since examples in Supplementary Fig. 4i–l aim to illustrate the shape of the current-voltage relationships, we removed the curves recorded in the presence of THC and left the control curves only, thus avoiding misinterpretation of the extent of current inhibition that can be substantially affected by the leak current amplitude.

It is unfortunate that basal calcium signals (non-transfected cells) in the studies of L490W are much higher than in studies of the other mutants (Source data file), perhaps further complicating the analysis of L490 in THC binding?

Yes, we agree. We have now emphasized this in the text and mentioned that the results of experiments with L490W have to be interpreted with caution (lines 200-202). We also have performed additional experiments with L490W and illustrated the new results in Supplementary Figure 4m-o (see above).

Other concerns:

In the revised manuscript, fluorescence data (control) are presented after subtraction of the non-specific changes in fluorescence for non-transfected cells caused by THC (lines 506-507). Is this also the case for experiments in Suppl. Fig. 4m,n? No controls are presented in Source data file.

No, we have not made control subtractions for these measurements because we did not observe non-specific effects caused by Ca^{2+} or econazole in the range of used concentrations.

The slope of the concentration-response curve to econazole is different in M497A compared to other mutants (Suppl. Fig. 4n). What could be the reason?

Similar to L490, M497 is located on the opposite site of the S5 helix, in which the W495 side chain makes direct contact with the econazole molecule. It is possible that due to its close location to the econazole binding site, the M497A mutation weakens the cooperativity that normally exists between four econazole binding sites (one per TRPV6 subunit). The corresponding possible explanation has been added to the text (lines 209–212).

Concentration of calcium is 10 mM (Suppl Fig. 4n)? Add to legend.

Done.

For electrophysiology and calcium measurements; define n and independent experiments (cell batches, transfections, replicates).

The corresponding information has been added to the figure legends.

Error bars indicate? Please add “data are represented as the mean \pm SEM/SD” or similar to legends (Figs 1 and 2, Suppl. Fig. 4). Software for calculation of mean, SD/SEM, IC50?

We have added the information about errors to the figure legends and mentioned in the Methods section that data analysis was performed using the computer program Origin 9.1.0 (lines 532-533, 545).

Voltage ramps: -120 to +120 mV (Fig. 1 legend, results and methods) or -100 to +70 mV (Fig. 1a and Suppl. Fig. 4i-l)? Number of experiments/cells (Suppl. Fig. 4i-l)? Can IC50 values be calculated as in Fig. 1a? $H_p = 0$ or -60 mV (see methods).

Thank you for noticing these inconsistencies. All ramps were from -100 to +70 mV and we have now corrected the corresponding statements in the text and figure legends.

The same data for calcium measurements are used in Fig. 1c and Fig. 2g. This should be stated within legends.

Done.

Suppl. Fig. 4n; Source data file does not support n = 12 for WT, rather n = 9.

Thank you for noticing. This has been fixed.

Arrow direction should be opposite in rotational views (Fig. 1d-f and Fig. 2a-d)?

Reviewer #4 is right. Thank you for noticing! The arrows have been reversed.

Reviewer #5 (Remarks to the Author):

I am satisfied with the revision of the manuscript. The additional simulations address properly the

concerns that I raised and provide further support to the conclusions. I now recommend publication of the manuscript.

We thank Reviewer #5 for the favorable assessment of our work.

REVIEWERS' COMMENTS

Reviewer #1 (Remarks to the Author):

The authors have improved the functional data in this paper, providing better support for some of their interpretations.

Reviewer #4 (Remarks to the Author):

The authors now give a more complete description and analysis of functional data that will help the journal's broad audience to better understand and evaluate the interesting findings of this study.